# Gradients in the biophysical properties of neonatal auditory neurons align with synaptic contact position and the intensity coding map of inner hair cells

Alexander L Markowitz[1,2], Radha Kalluri[1,2,3]*

[1]Neuroscience Graduate Program, University of Southern California, Los Angeles, United States; [2]Department of Otolaryngology, Keck School of Medicine, University of Southern California, Los Angeles, United States; [3]Zilkha Neurogenetic Institute, Keck School of Medicine, University of Southern California, Los Angeles, United States

**Abstract** Sound intensity is encoded by auditory neuron subgroups that differ in thresholds and spontaneous rates. Whether variations in neuronal biophysics contributes to this functional diversity is unknown. Because intensity thresholds correlate with synaptic position on sensory hair cells, we combined patch clamping with fiber labeling in semi-intact cochlear preparations in neonatal rats from both sexes. The biophysical properties of auditory neurons vary in a striking spatial gradient with synaptic position. Neurons with high thresholds to injected currents contact hair cells at synaptic positions where neurons with high thresholds to sound-intensity are found in vivo. Alignment between in vitro and in vivo thresholds suggests that biophysical variability contributes to intensity coding. Biophysical gradients were evident at all ages examined, indicating that cell diversity emerges in early post-natal development and persists even after continued maturation. This stability enabled a remarkably successful model for predicting synaptic position based solely on biophysical properties.

*For correspondence:
radha@usc.edu

## Introduction

The bipolar afferent neurons of the auditory system are the principle conduits for information transfer from the sensory periphery to the brainstem. In mature mammals, one inner hair cell provides input to type I spiral ganglion neurons (SGN) with different response properties (*Liberman, 1982*). Some SGN have high rates of spontaneous discharge, low-intensity thresholds and encode sound over a limited range of intensities (high-SR group). Other SGN have low spontaneous rates and higher intensity thresholds (low-SR group). Together, these *spontaneous rate groups* (SR groups) convey the vast range of sound intensities needed for normal hearing.

Despite their fundamental importance to sound encoding, the biophysical mechanisms defining sensitivity to sound intensity remain unknown. Decades of research focusing on this question have led to multiple classification schemes based on in vivo physiology and active zone morphology (*Kawase and Liberman, 1992*; *Liberman and Dodds, 1984*; *Merchan-Perez and Liberman, 1996*). Specifically, these studies report an association between synaptic position on inner hair cells and intensity thresholds; wherein high-threshold, low-SR SGN preferentially synapse on the modiolar face of an inner hair cell, and low-threshold high-SR SGN synapse on the pillar face.

Several anatomical and physiological features are correlated to synaptic position. These include differences in the type, density, and voltage dependence of pre-synaptic $Ca^{2+}$ channels and $Ca^{2+}$ sensors (*Ohn et al., 2016*; *Wong et al., 2014*), the relative complexity of the synaptic ribbon

(reviewed in *Moser et al., 2006*; *Safieddine et al., 2012*) and the expression of post-synaptic gluta-mate receptors (*Liberman et al., 2011*).

Many of the correlations between anatomical features and afferent response features are counter-intuitive and inconsistent with expectations based on other systems. For example, the pre-synaptic active zones opposing high-SR SGN have smaller ribbons (*Merchan-Perez and Liberman, 1996*) and calcium currents (*Ohn et al., 2016*) than those opposing low-SR SGN. This stands in contrast to large ribbons generating faster excitatory post-synaptic current (EPSC) rates in retinal ganglion cells (*Mehta et al., 2013*). Whether heterogeneity in ribbon morphology produces differences in average EPSC rates and heterogeneity in EPSC amplitude and kinetics at inner hair cell synapses (for example *Grant et al., 2010*) remains to be determined. In summary, the factors responsible for defining each SR-subgroup and the diversity of their responses to sound intensity remain poorly understood.

Here, we ask whether cell-intrinsic diversity among SGN contributes to differences in sound-inten-sity coding. Previous studies in cultured spiral ganglion explants established that SGN are rich in their complements of ion channels and respond to injected currents with diverse firing patterns (*Mo and Davis, 1997*; *Davis, 2003*; *Liu et al., 2014a*). Systematic variation of somatic ion channels along functionally relevant spatial axes would suggest that such variation is relevant for neuronal computations. For example, a previous study using semi-intact cochlear preparations reported that type I SGN, which contact inner hair cells and are the primary conduits for sensory information, can be biophysically distinguished from type II SGNs, which contact the electromotile outer hair cells, by the kinetics of their potassium channels (*Jagger and Housley, 2003*). Single-cell RNA-sequencing studies report that type I SGN can be further divided into genotypic subgroups based on RNA expression levels for a variety of proteins including ion channels, calcium-binding proteins and pro-teins affecting $Ca^{2+}$ influx sensitivity (*Shrestha et al., 2018*; *Sun et al., 2018*; *Petitpré et al., 2018*; *Sherrill et al., 2019*). However, no study has tested whether differences in type I SGN intrinsic bio-physical properties are logically aligned with the in vivo SGN groups (i.e. SR-groups).

Here, we used simultaneous whole-cell patch clamping and single-cell labeling of SGN in acute semi-intact cochlear preparations. By keeping hair cells and neurons connected, this preparation allows tests for links between post-synaptic cellular biophysics and putative SR-subgroups (as inferred from synaptic location). Our data show strong correlations between SGN biophysics and synaptic position throughout the first two weeks of post-natal development. This is the first study to demonstrate such an alignment between the intrinsic properties of spiral ganglion neurons and their function. Moreover, the strength of the correlations enabled us to predict synaptic position based solely on easily measured biophysical properties, suggesting a simple route to identify putative SR-subtypes in SGN.

## Results

### Developing spiral ganglion neurons can be classified based on their pattern of connectivity with hair cells

We characterized the morphology and electrophysiology of spiral ganglion neurons (SGN) using whole cell patch-clamp recordings combined with single-cell labeling from the middle turn of rat cochleae ranging in age from post-natal day (P) 1 through P16. Labeled SGN were classified in the XZ plane of the organ of Corti, where both sides of the inner hair cell, the tunnel of Corti, and three rows of outer hair cells (OHCs) are identifiably organized for classification (schematized in *Figure 1A*). The dendritic morphology of individual SGN was characterized by introducing biocytin into the soma via the patch pipette (*Figure 1B &C*). The biocytin diffused to fill the peripheral termi-nal (*Figure 1B*). The biocytin filled neuron was visualized after fixation and immuno-processing using fluorophore conjugated streptavidin (*Figure 1B*; see Materials and methods).

We identified two subtypes of SGN based on their connectivity to cochlear hair cells. Type I SGNs contacted inner hair cells (*Figure 1D–F*; n = 50), while type II SGNs (*Figure 1G*; n = 9) pro-jected underneath the inner hair cells, turned toward the cochlear base and made multiple contacts under the outer hair cells (ranging from 6 to 15 contacts). Of the type I afferents, many reconstructed afferent fibers had terminal branches that did not touch on any hair cells but projected into neigh-boring areas, proximal to the hair cells (*Figure 1D* .2, 1E.2, 1F.2). These fibers project on areas that

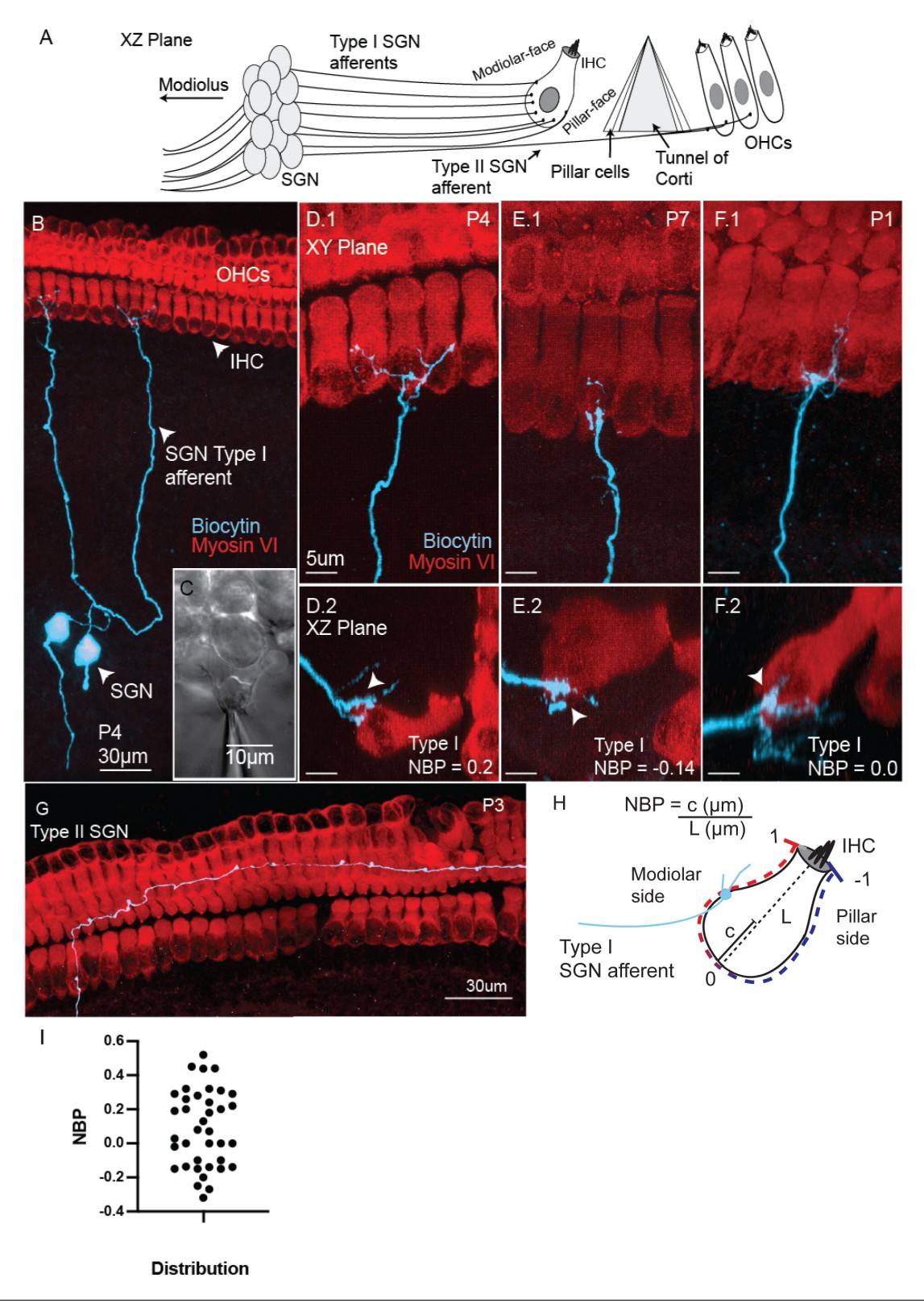

**Figure 1.** Semi-intact preparation combines SGN biophysics with identification of pattern of connectivity with hair cells. (**A**) Schematic of the innervation at auditory hair cell along with key anatomical structures provided with the semi-intact preparation presented in the XZ plane. (**B**) Confocal image of two type I spiral ganglion neurons (SGN) injected with biocytin in the acute semi-intact preparation of rat middle-turn cochlea. Hair-cells and biocytin-labeled fibers are visualized by immunolabeling for Myosin VI (red; hair cells) and streptavidin conjugated to Alexa Flour 488 (green, fiber). (**C**)
*Figure 1 continued on next page*

*Figure 1 continued*

Myelinating Schwann cells are mechanically removed by suction pipettes to make SGN somata accessible to recording electrodes. D.1-F.1. Confocal images in the XY plane show type I SGN afferent fibers making exclusive connections with inner hair cells. D.2-F.2 The synaptic position of the afferent fiber onto the inner hair cell is assessed by examining the cross-section of the hair cell in XZ plane. (G) Type II SGN projecting radially and turn within the outer hair cell region. (H) A schematic of an individual inner hair cell with guides showing how the normalized basal position (NBP) is measured for each type I SGN. NBP is measured by the synaptic position along the radial axis of the hair cell,c, divided by the maximum length of the inner hair cell, L. Positive NBP values indicate that the synaptic position is on the modiolar-side of the inner hair cell, while negative NBP values indicate that the synaptic position is on the pillar-side of the inner hair cell. (I) The distribution of type I SGN contacts as a function of NBP. NBP values averaged at 0.11 +/- 0.23 with a range from −0.27 to +0.52 (n = 38).

The online version of this article includes the following figure supplement(s) for figure 1:

**Figure supplement 1.** Polarization vector shows that branching fibers have a preferred polarization to either modiolar or pillar face.

are not visualized with Myosin VI signaling. In other cases (n = 2), type I SGN terminals extend to a neighboring inner hair cell. Finally, in one case, the fiber projected into the outer hair cell layer but did not turn. This fiber was also negative for peripherin (see Materials and methods). We did not encounter any other examples of putative Type I fibers extending out to the OHC region such as observed in other studies (e.g. *Huang et al., 2007*). However, because peripherin is not a definitive marker for Type II fibers, especially in early development, we excluded this fiber from further analysis.

In this age range, the branching patterns of Type I fibers are highly polarized to either the modiolar or pillar face of a hair cell. For those fibers that contact the modiolar face, nearly all the branches of the fiber are also located on the modiolar side of the hair cell and vice-versa (See *Figure 1—figure supplement 1* and Appendix 1). Whether this indicates the presence of selective guidance cues for modiolar versus pillar-contacting fibers remains to be tested. Although the polarization of fibers allowed us to confidently assign fibers as being either modiolar- or pillar-contacting, we moved away from this bimodal classification in preference for a more continuous scale as described below.

We defined a continuous position scale, which we refer to as 'Normalized Basal Position' (NBP), to quantify where SGN terminals contact the inner hair cells (*Figure 1H*). The 'modiolar' and 'pillar' halves of the hair cell were first defined by drawing a bisecting plane from the basal pole to the cuticular plate of the hair cell in the XZ plane of the confocal scan. The origin (NBP = 0) was set at the basal pole of the bisected inner hair cell. To determine NBP we first computed the fractional distance of the contact position along the radial axis of the hair cell ($c$) relative to the length of the hair cell ($L$) (see *Figure 1H*). The normalization by hair cell length standardized the position scale across preparations and controlled for possible changes in hair cell length with age. This value was multiplied by +1 if the contact position was on the modiolar face and by −1 if the contact position was on the pillar face.

NBP measurements were only made when the preparation was in relatively pristine condition after electrophysiological manipulations and immunostaining. Therefore, only a subset of the data had this measurement (n = 38 of 50). If there were multiple branches that contacted the inner hair cell, we used the most extreme contact position relative to the basal pole for our analyses. NBP values ranged from −0.32 to +0.52 (n = 38) (*Figure 1I*). In the preparations where we could not measure the NBP, we could classify the position on a bimodal system as either modiolar-contacting (NBP >0) or pillar-contacting (NBP <0). In total, we classified 34 fibers as modiolar-contacting, 16 fibers as pillar-contacting, and nine as type II SGN. The non-uniform distribution between Type I, Type II and subtypes of Type I fibers is in line with the pattern of innervation previously reported in rat and cat animal models, where the vast majority of SGN fibers appear to land on the modiolar-face of the inner hair cell (*Liberman and Oliver, 1984*; *Kalluri and Monges-Hernandez, 2017*).

## Biophysical properties of spiral ganglion neurons change with normalized basal position

### Diversity of firing patterns in current clamp

In current clamp mode, we measured the firing patterns evoked from SGN somata in response to 1000 ms long injected current steps (*Figure 2*). Cells maintained their natural resting potential before we injected current. Firing patterns were diverse and qualitatively binned into four groups

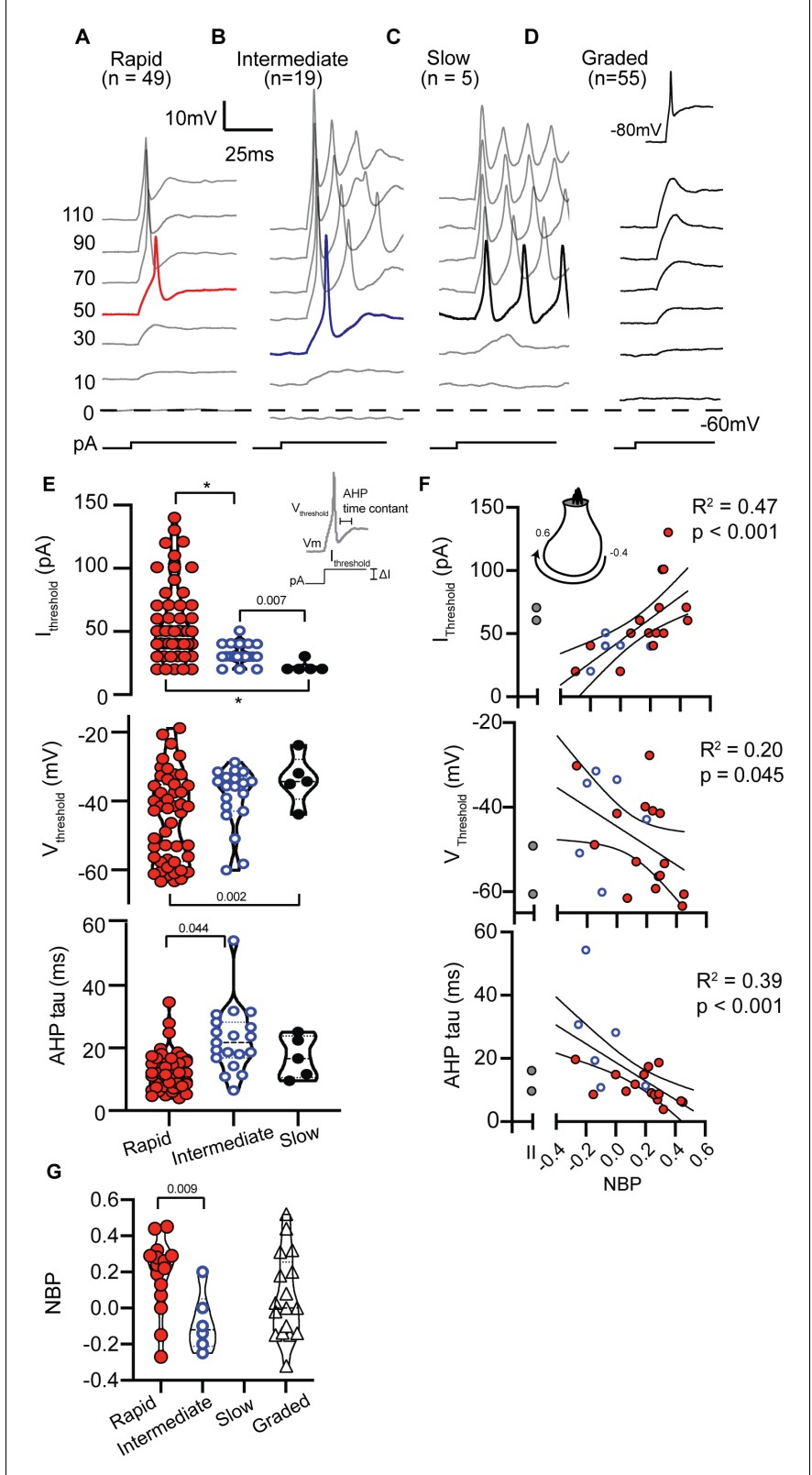

**Figure 2.** SGN can be qualitatively grouped by firing patterns in response to step current injections. (A) Rapidly-accommodating neurons fire a single action potential. (B) Intermediately-accommodating neurons fire more than one action potential and eventually reach a steady-state potential. (C) Slowly-accommodating neurons fire multiple actions and do not reach a steady-state potential. (D) Graded-response neurons do not produce an action

*Figure 2 continued on next page*

*Figure 2 continued*

potential, but produce graded-depolarization in response to incrementing current steps. These neurons are capable of firing action potentials after their membrane potential are held at −80 mV (inset). (E) Action potential features including current threshold ($I_{threshold}$), voltage threshold ($V_{threshold}$), and after hyperpolarization potential time constant (AHP tau) vary among spiking neurons. The p-values indicate when two means were assessed to be significantly different (see Materials and methods); asterisks indicate p<0.001. (F) Action potential features were significantly correlated with normalized basal position (NBP). As NBP values increased, current thresholds increased (r(20) = 0.68, p=0.0007), voltage thresholds hyperpolarized (r(20) = −0.45, p=0.045), and AHP time constants became faster (r(20) = 0.62, p<0.0001). Best fit lines with 95% confidence intervals are plotted to show the gradients along the normalized basal position axis. (G) The distribution of firing pattern subgroups as a function of NBP shows a significant relationship between the firing-pattern subgroup and NBP (p=0.0097, t-test), with rapidly-accommodating neurons found primarily on the modiolar-face and intermediately-accommodating neurons found on the pillar-face of the inner hair cell. Graded-firing neurons are found throughout the NBP scale on both the modiolar and pillar faces of the hair cell.

The online version of this article includes the following source data for figure 2:

**Source data 1.** Current-clamp features versus normalized basal position data set for *Figure 2F*.

based on degree of accommodation. *Rapidly-accommodating* firing patterns had a single action potential at the onset of a step of current (*Figure 2A*). *Intermediate-accommodating* firing patterns had more than one action potential (*Figure 2B*) but accommodated more quickly than *slowly-accommodating* firing patterns which typically had multiple action potentials throughout the duration of the stimulus (*Figure 2C*). The fourth response category did not have clearly identified action potentials and instead had graded depolarizations in response to a family of current injections (*Figure 2D*). Although neurons with *graded-responses* did not fire action potentials when perturbed from their natural resting potential, they were capable of firing action potentials when they were first artificially held at −80 mV (*Figure 2D* inset); the more negative holding potential presumably relieves sodium channel inactivation.

The four patterns were not encountered in equal proportion even though recording position was varied to minimize sampling bias (see Materials and methods). Rapidly accommodating firing patterns were the most prevalent (n = 49), followed by intermediate-accommodating firing patterns (n = 19). Slow-accommodating patterns were the least prevalent (n = 5), with all observations occurring between P1 and P3, suggesting that they might be an immature phenotype in acute preparations. Graded-responses were observed at all ages (n = 55). All firing patterns were observed in Type I SGN, while only rapidly-accommodating (n = 2; P8) and graded-responses (n = 7; P1-P7) were observed in Type II SGN. Therefore, we did not find a simple association between firing pattern and Type I and Type II fiber morphology.

To move beyond a qualitative classification that forces responses into discrete firing pattern groups, we measured an array of response features to examine subtle differences. Our reasoning was that variability in detailed response features also reflects variability in the intrinsic biophysical properties of a neuron, including ion channel composition (*Liu et al., 2014a*; *Liu and Davis, 2014b*). This shift toward quantification was especially important because rapidly-firing, intermediate-firing and graded-responses were not always easily discriminable, and slowly-accommodating firing patterns were rarely encountered.

In spiking neurons (i.e. rapidly-, intermediate- and slowly-accommodating), we measured features that are directly related to an action potential (*Figure 2E* inset). Current thresholds, voltage thresholds and after hyperpolarization time-constants varied by spike-pattern (F(2,22.8) = 29.3, p<0.001; F(2,13.0) = 5.06, p=0.024; F(2,10.3) = 8.14, p=0.008, respectively by Welch's ANOVA). Overall, the action potential features were broadly distributed within each firing group (see *Figure 2E*). Although there were no significant differences in resting potential among the three spiking neurons (potential measured in the absence of current injection; F(2,10.43) = 2.24, p=0.15, Welch's ANOVA), spiking-neurons as a group had more hyperpolarized resting potential than the non-spiking graded-responding neurons (−56.86 +/- 0.98 mV vs. −51.77 +/- 1.14 mV, respectively; p=0.0010, t-test). A depolarized resting potential may account for why the graded-response neurons spiked only after first being hyperpolarized to −80 mV (*Figure 2D*, inset). The hyperpolarization presumably relieves sodium channel inactivation.

## Current-clamp features of spiking-neurons are correlated with normalized basal position

Next we tested for and found significant correlations between the action potential features of spiking-neurons and normalized basal position (NBP) (*Figure 2F*). Current-clamp features of Type II neurons are shown as gray symbols to the left of each correlation plot but are not included in the regression analysis. Larger currents were needed to initiate spiking as contact positions moved from the pillar face (NBP <= 0) to the modiolar face (NBP >0) (r(20) = 0.68, p=0.0007). Voltage thresholds hyperpolarized (r(20) = −0.45, p=0.045), resting potentials hyperpolarized (r(20) = −0.45, p=0.01) and after-hyperpolarization times became shorter with increasing values of NBP (r(20) = -0.62, p<0.0001). Note that although voltage thresholds were significantly correlated with NBP, the relative

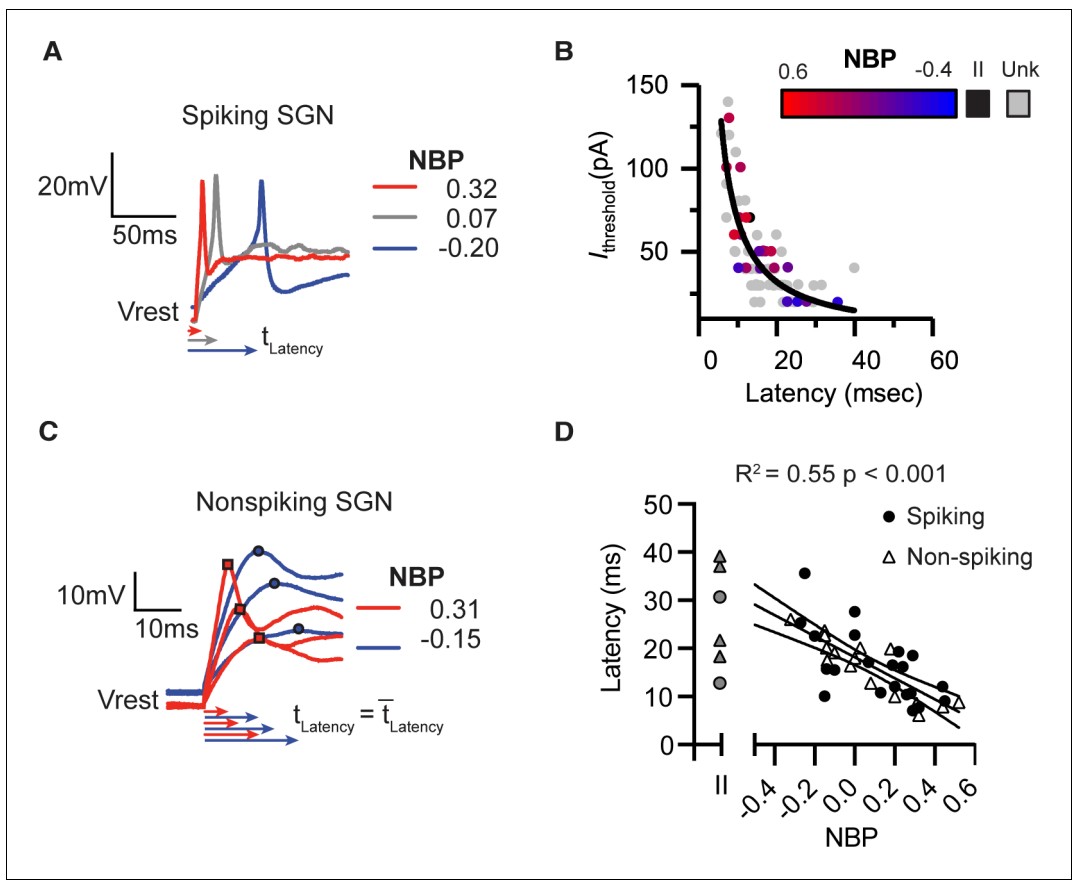

**Figure 3.** Response latencies are alternate predictors for normalized basal position. There is a significant inverse relationship between first-spike latency and current thresholds. Black line shows a fit $I_{thresh}$ = constant/latency. Color transition from red to blue indicates position on NBP scale. Type II SGN are colored black. Unlabeled and therefore unclassified SGN are in gray. (**A**) First spike latencies compared between three spiking SGN with different NBP values. Latencies are faster for more positive NBP (**B**) In non-spiking neurons, response latency is computed by averaging the time from the onset of the stimulus until peak depolarization potential over a series of current steps. Similar to the dependence of first-spike latency, average response latency is faster for modiolar-contacting SGN (NBP >0, red) than for pillar-contacting SGN (NBP <0, blue). (**C**) Response latencies for all spiking (black dot) and non-spiking (triangle) neurons plotted against NBP. In both spiking- and non-spiking SGN, response latencies become faster as NBP values increase ($R^2$ = 0.55, p<0.001). Latencies are highly variable across type II fibers (gray symbols).

The online version of this article includes the following source data for figure 3:

**Source data 1.** Current threshold versus latency dataset for *Figure 3B*.
**Source data 2.** Latency versus normalized basal position dataset for *Figure 3D*.

threshold (change in voltage relative to the resting potential) was not significantly correlated with NBP (r(20) = 0.29, p=0.2).

The correlations between individual action potential features and basal position are like the gross differences between rapidly- and intermediate-accommodating spike patterns. Intermediately accommodating neurons (blue circles), with their tendency for smaller current thresholds and slow AHP time courses, were more prevalent on the pillar-side of the inner hair cell while rapidly-accommodating neurons (red circles) were found throughout the NBP scale, but were more prevalent on the modiolar face (*Figure 2G*). The average NBP for intermediately-accommodating neurons is more negative than that for rapidly-accommodating neurons (−0.08 +/- 0.7 vs. 0.18 +/- 0.5, p=0.0097, test-t).

## Current-clamp features of non-spiking neurons are also correlated with normalized basal position

In spiking neurons, we showed that current threshold was significantly correlated with basal position, but non-spiking neurons with their graded responses did not have thresholds. Given that many labeled neurons had graded responses, we examined other current-clamp features not related to spiking and found that response latencies of spiking- and non-spiking neurons (described below) were also significantly correlated with basal position. This ultimately allowed us to pool the spiking- and non-spiking neurons into one group for subsequent analysis.

In spiking neurons, we measured response latency as the time delay between the onset of the current injection and the first spike induced by threshold-level current steps (first-spike latency, $t_{Latency}$). First-spike latencies were shorter in a representative modiolar-contacting neuron than in a pillar-contacting neuron (*Figure 3A*). We noted a strong covariation between current threshold and latency, with latency decreasing as current thresholds increased (*Figure 3B*). This relationship is well fit by the following equation ,

$$I_{thresh} \, \alpha \, 1/t_{Latency} \tag{1}$$

The form of the relationship between current threshold and latency can be understood when a neuron is viewed as a circuit consisting of a resistor and capacitor in parallel (as a cell membrane is often modeled). Based on Ohm's law, we expect that the amount of current needed to effect a fixed change in voltage increases linearly with input conductance. Similarly, the time it takes for the voltage to change (membrane time constant) is inversely related to the conductance. Thus, here the covariation between latency and current threshold is related by their likely common dependence on the underlying conductance.

In non-spiking neurons which don't have current thresholds, we measured response latency as the average time needed to reach peak depolarization for a family of current steps. This is shown for two example neurons in *Figure 3C*. The depolarization of the modiolar-contacting neuron reached peak potential earlier than did that of the pillar-contacting neuron. The response latencies of graded/non-spiking and spiking-neurons are faster for fiber-contact positions closer to themodiolar face of the inner hair cell (r(34) = −0.74, p<0.0001, pooled; r(20)=-0.65, p=0.0013, spiking; r(14)=-0.89, p<0.0001, non-spiking). Analysis of co-variance confirmed that the mean latency was not significantly different between spiking and non-spiking groups (p=0.94, ANCOVA,*Table 1* ). Nor was the slope of the regression significantly different between spiking and non-spiking neurons (p=0.71, ANCOVA, *Table 1*). This allowed us to pool the two groups for subsequent analysis (*Figure 3C& D*).

## Diversity in whole-cell currents in voltage-clamp

Next, we discuss the dependence of whole-cell currents (measured in voltage-clamp) on normalized basal position. In *Figure 4A–C*, we show the currents measured in response to a family of voltage steps (from −120 mV to +70 mV) in three example somata; each identified as a modiolar-contacting, pillar-contacting, and type II spiral-ganglion fiber, respectively. The magnitude of whole-cell currents was highly variable. We did not correct for series resistance errors but excluded the possibility that this influences our interpretation of the results (see Appendix 2 and *Figure 4—figure supplement 1*). Each response consisted of net inward (negative) and outward (positive) currents. The inward currents were largely transient, activating and inactivating quickly, consistent with our expectations for

**Table 1.** Analysis of covariance.

| Figure | Source | d.f. | Sum of squares | F-score | p-value |
|---|---|---|---|---|---|
| 3D Latency | Spike (Spike/Non-Spike) | 1 | 0.158 | 0.0052 | 0.94 |
| | NBP | 1 | 967.79 | 32.04 | <0.0001 |
| | Spike * NBP | 1 | 4.14 | 0.13 | 0.71 |
| 4F $g_{max}$ | Spike | 1 | 150.99 | 1.37 | 0.25 |
| | NBP | 1 | 1578.89 | 14.29 | 0.0006 |
| | Spike * NBP | 1 | 0.038 | 0.0004 | 0.98 |
| 4 G $g_{-30}$ | Spike | 1 | 41.27 | 1.97 | 0.17 |
| | NBP | 1 | 568.15 | 27.16 | <0.0001 |
| | Spike * NBP | 1 | 12.14 | 0.58 | 0.45 |
| 4 H $V_{1/2}$ | Spike | 1 | 632.64 | 3.53 | 0.068 |
| | NBP | 1 | 743.44 | 4.15 | 0.0495 |
| | Spike * NBP | 1 | 107.86 | 0.60 | 0.44 |
| 4I $\tau_{inact}$ | Spike | 1 | 18715.3 | 3.64 | 0.068 |
| | NBP | 1 | 42318.4 | 8.24 | 0.0084 |
| | Spike * NBP | 1 | 9.85 | 0.0019 | 0.96 |
| 6C. $g_{max}$ | Class (Modiolar/Pillar) | 1 | 1508.38 | 7.32 | 0.0107 |
| | Age | 1 | 1963.45 | 9.53 | 0.0041 |
| | Class* Age | 1 | 1232.78 | 5.98 | 0.0199 |
| 6E $I_{threshold}$ | Class (M/P) | 1 | 3827.39 | 8.87 | 0.0087 |
| | Age | 1 | 2333.04 | 5.35 | 0.0335 |
| | Class * Age | 1 | 280.75 | 0.64 | 0.4334 |
| 6G Latency | Class (M/P) | 1 | 577.732 | 36.97 | <0.0001 |
| | Age | 1 | 483.887 | 30.96 | <0.0001 |
| | Class * Age | 1 | 131.247 | 8.4 | 0.0066 |

currents conducted by sodium channels. The outward currents inactivated relatively slowly to produce a nearly steady net current around 400 ms after the onset of the current step.

Previous studies have shown that SGN have both high-voltage- and low-voltage-gated potassium currents in diverse amounts which can lead to differences in thresholds and excitability (*Liu et al., 2014a*; *Lv et al., 2010*). Here, we assume that the steady-state outward currents we measure in voltage-clamp are largely dominated by potassium currents. To quantify the size and voltage dependence of these currents, we converted them into approximate whole-cell conductance by dividing the steady-state current ($I_{ss}$) by the driving force for potassium ($V_m$-$E_K$), where $V_m$ is the membrane voltage and $E_K$ is the reversal potential for potassium for our recording conditions. The resulting whole-cell conductance curves were fit with Boltzmann curves from which we characterized three features: 1) the value at -30 mV ($g_{-30}$) as a rough gauge for the size of low-voltage gated potassium conductances, 2) the maximum value ($g_{max}$) to gauge total potassium conductance, and 3) the voltage for half-maximum activation ($V_{1/2}$) to gauge the relative balance between low-voltage and high-

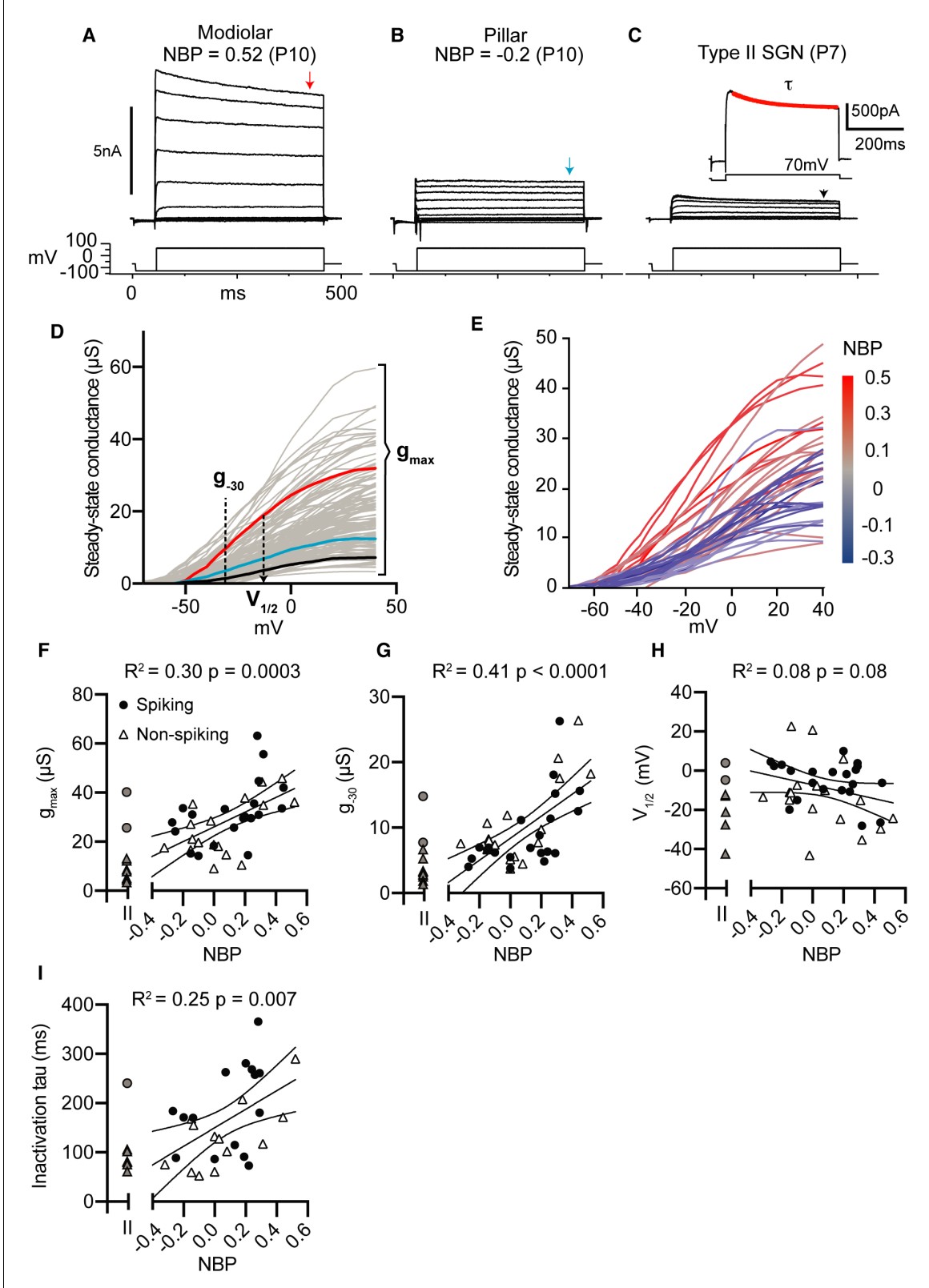

**Figure 4.** Net whole-cell conductances increase as SGN synaptic position moves from pillar to modiolar-face of the inner hair cell. (A–C) Voltage-clamp responses of a modiolar-contacting (A), pillar-contacting type I (B), and a type II SGN (C). For each cell, we held the potential at −60 mV, followed by a series of voltage 400 ms steps from −120 mV to 70 mV, and then finally back to the holding potential. For each cell, we measured current-voltage relationships at approximate steady-state (400 ms indicated by the arrow). We measured the time course for outward current inactivation by fitting a
*Figure 4 continued on next page*

*Figure 4 continued*

single exponential curve to the current evoked by a +70 mV voltage step (inset). (**D**) Steady-state conductance in microSiemens as a function of command voltage of all SGN recorded with three measurements: the magnitude of conductance at −30 mV ($g_{-30}$), the maximum conductance ($g_{max}$), and the half-activation potential ($V_{1/2}$) via Boltzmann function. (**E**) Steady-state conductances of identified type I SGN as a function of normalized basal position. (**F–I**) Voltage clamp properties as a function of normalized basal position (NBP) (Spiking, black dot; non-spiking, triangle). Linear regression models were fitted to each parameter with statistical significance test of p<0.05. All parameters were significant except for $V_{1/2}$.

The online version of this article includes the following source data and figure supplement(s) for figure 4:

**Source data 1.** Voltage-clamp features versus normalized basal position for *Figure 4F* through 4I.
**Figure supplement 1.** Considering the influence of space-clamp and series resistance errors.

voltage conductance. In several cases (n=10 out of 128), the conductance curves did not saturate. These SGN were omitted from subsequent analyses. In addition to characterizing the voltage dependence and size of the steady-state outward conductance, we measured the time constant over which the outward current inactivates at +70 mV, referred to here on as $\tau_{inact}$ (*Figure 4C*, inset).

## Whole-cell conductances grow as a function of normalized basal position

As we showed in *Figures 2* and *3*, current thresholds and response latencies were strongly correlated with normalized basal position. We hypothesized that voltage-clamp responses of whole-cell currents would be similarly correlated with basal position. Indeed, we can see qualitatively that modiolar-contacting neurons have larger maximum whole-cell conductance compared to pillar-contacting neurons (as indicated by the color transition from blue to red as normalized basal position becomes more positive (*Figure 4E*).

Individual conductance curve features show significant correlations with normalized basal position. Maximum whole-cell conductance ($g_{max}$) increased as a function of NBP (r(34) = 0.57, p=0.0003) (*Figure 4F*). Neurons terminating on the modiolar-face also had large $g_{-30}$, suggesting that they may have had larger low-voltage gated components (r(34) = 0.64, p<0.0001) (*Figure 4G*). However, $V_{1/2}$ was not significantly correlated to basal position (r(34) = −0.23, p=0.17) (*Figure 4H*). Inactivation time constant was correlated with normalized basal position; with the largest time constants found for neurons on the modiolar face of the inner hair cell (r(25) = 0.50, p=0.0087). Note that the age dependence of all the voltage-clamp features was similar between non-spiking and spiking cells (evaluated by a two-way ANCOVA, p>0.05 for all features reported in *Figure 4*) (*Table 1*). For comparison, the voltage-clamp features for Type II neurons are shown in gray symbols to left of each correlation plot, these cells were not included in the prediction models presented next.

## Current- and voltage-clamp features together predict normalized basal position

As we showed in the previous section, we found significant correlations between several biophysical properties and normalized basal position. In this section, we generate prediction models for normalized basal position based on a linear combination of multiple biophysical features. The full details of the model building can be found in the methods. Briefly, the process began by pooling all the current-clamp and voltage-clamp features that are significantly correlated with basal position. To prevent overfitting, the number of variables retained into the model was reduced using a backward stepping process, with highly collinear (i.e. redundant) variables removed. By analyzing the collinearity between biophysical properties, the model building process also pointed at the underlying relationships between variables. In *Figure 5*, we show four prediction models corresponding to the four ways in which the data were divided in different subsets. The first model is based only on the subset of neurons that spiked in response to current steps. These neurons were pooled across a wide age range (from P3-P10). The second model is over the same age range but also includes non-spiking neurons. By including non-spiking neurons, the model limits the current-clamp variables to those that can be measured in the absence of action potentials. The third and fourth models include spiking and non-spiking neurons but filter the data into narrow age bins. By looking at narrow age ranges, we test if spatial gradients are present at multiple developmental time points.

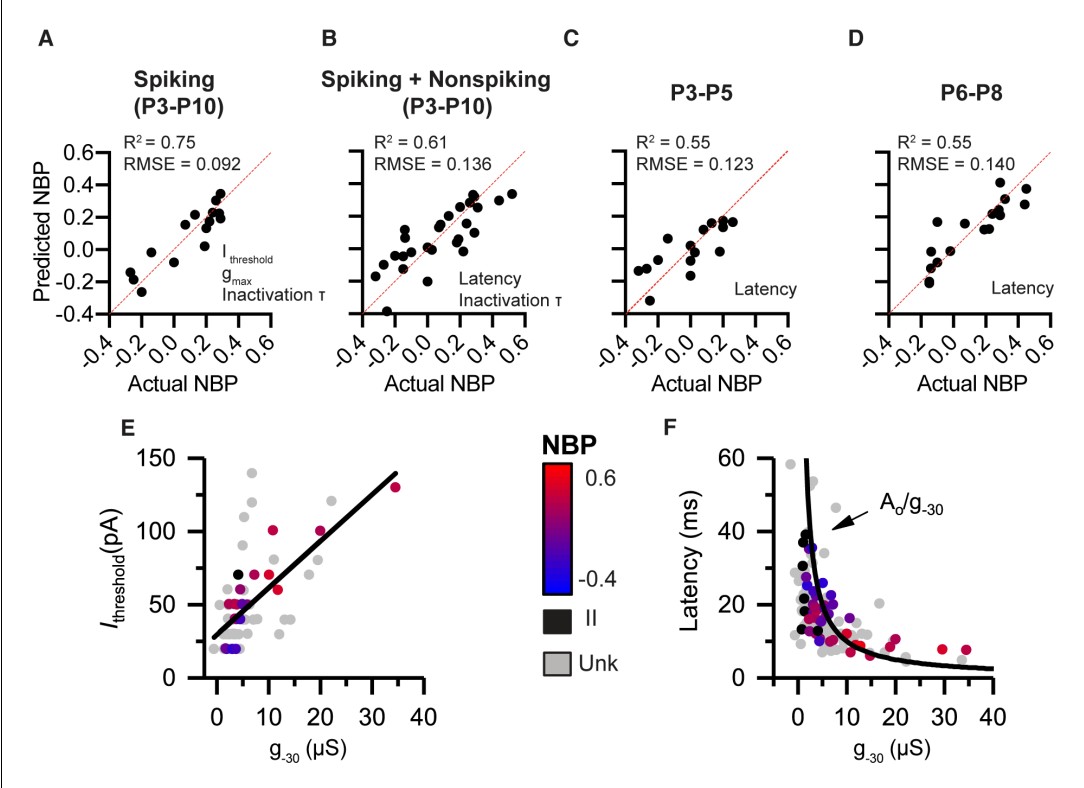

**Figure 5.** Combining voltage-clamp and current-clamp features in multiple-variable regression models to predict normalized basal position (NBP) based on biophysics. (**A**) Prediction model for spiking neurons (P3–P10) relies on $I_{thresh}$, $g_{max}$ and $t_{inact}$. (**B**) Prediction model that pools spiking and non-spiking neurons (P3–P10) is successful by using latency in place of current threshold (**C,D**), respectively. Biophysical gradients are still successful at predicting spatial position even after filtering data into narrow age bins. P3-P5 in C and P6-P8 in D. Predictor variables used in each model are presented at the bottom right of the predicted vs. actual plot. (**E**) Current threshold increase with increases in $g_{-30}$. (**F**) Response latency as a function of $g_{-30}$. Fit is a function of the form latency = constant/$g_{-30}$.

## Model 1: Spiking neurons only (P3-P10)

The first multiple regression model was on spiking-neurons between P3 and P10. The following seven current- and voltage-clamp features were significantly correlated with normalized basal position: current threshold, voltage threshold, first-spike latency, AHP time constant, resting potential, $g_{-30}$, $g_{max}$, and $\tau_{inact}$ (**Table 2**, see also **Figure 2**). After reducing the number of input variables by accounting for variable inflation and via the backward stepping process, the best prediction model for spiking-neurons included three variables: current threshold, $g_{max}$, and $\tau_{inact}$ (**Figure 5A**). Current-threshold was the most significant variable predicting the normalized position (recall that it accounted for nearly 47% of the variance on its own, **Figure 2F**). This was followed by $g_{max}$ and $\tau_{inact}$. Note that other variables like latency and $g_{-30}$ were individually more significantly correlated with basal position than $g_{max}$ or $\tau_{inact}$, but they were strongly collinear (or redundant) with current threshold and thus were eliminated during variable reduction.

The performance of the prediction model is illustrated by plotting predicted basal position against actual position (**Figure 5A**). If the model were perfectly accurate, each point (representing an individual neuron) would lie on the 45-degree trend line. The more spread the points are from the line, the less accurate the prediction. The model (**Table 3**) accounted for 75% of the variance in the data (adjusted- $R^2$ = 0.75, p=0.0004) and accurately predicted the basal position within approximately +/- 9% margin of error (RMSE = 0.092).

## Model 2: Spiking plus non-spiking neurons

In the absence of current threshold, the most influential predictor variable for non-spiking neurons is response latency. Note that although response-latency is strongly correlated with basal position for

**Table 2.** Correlations with normalized basal position.

| | | Spiking neurons P3-P10 | | Non-spiking P3-P10 | | Spiking plus non-spiking P3-P5 | | Spiking plus non-spiking P6-P8 | |
|---|---|---|---|---|---|---|---|---|---|
| | | R | p-value | R | p-value | R | p-value | R | p-value |
| *Current-clamp features* | *$I_{Threshold}$ | 0.68 | 0.0007 | | | 0.81 | 0.0080 | 0.44 | 0.0772 |
| | *First-spike Latency | −0.65 | 0.0013 | | | −0.76 | 0.0168 | −0.45 | 0.22 |
| | *Voltage Threshold | −0.39 | 0.0169 | | | −0.61 | 0.0837 | 0.15 | 0.7390 |
| | *AHP Tau | −0.61 | 0.0031 | | | −0.65 | 0.0217 | −0.21 | 0.5616 |
| | *AP Height | 0.31 | 0.17 | | | 0.03 | 0.5634 | 0.63 | 0.0069 |
| | Average Response Latency | | | −0.89 | <0.0001 | −0.76 | 0.0010 | −0.77 | 0.0003 |
| | Resting Potential | −0.54 | 0.0112 | −0.31 | 0.229 | −0.07 | 0.8048 | −0.48 | 0.0487 |
| Voltage-clamp features | $g_{max}$ | 0.52 | 0.0166 | 0.58 | 0.0177 | 0.78 | 0.8799 | 0.62 | 0.0083 |
| | $V_{1/2}$ | −0.27 | 0.233 | −0.39 | 0.137 | −0.10 | 0.71 | −0.09 | 0.74 |
| | $g_{-30}$ | 0.59 | 0.0045 | 0.74 | 0.0010 | 0.44 | 0.10 | 0.59 | 0.0122 |
| | $\tau_{inact}$ | 0.5 | 0.0066 | 0.67 | 0.0179 | 0.48 | 0.10 | 0.45 | 0.1459 |

both spiking- and non-spiking neurons, it is also highly correlated with current threshold (r(20) = -0.64, p < 0.0001) (*Figure 3B*). This collinearity between current-threshold and response latency was significant enough to inflate the models if both variables were included. Thus, none of the models use both current threshold and latency. The best model after pooling spiking- and non-spiking neurons used response latency and $\tau_{inact}$ to best predict NBP. This regression model accounts for 61% of the total variance (adjusted-$R^2$ = 0.61, p < 0.0001) and accurately predicts the basal position within a 13.6% margin of error (RMSE = 0.136) (*Figure 5B*).

## Models 3 and 4: Spatial gradients in biophysical properties are present in early post-natal days and maintained until the onset of hearing

Next we tested whether spatial gradients in biophysical properties were still present if the data were filtered into two narrow age ranges; P3-P5 and P6-P8. We ran the same model building procedure as in the previous sections. By binning spiking- and non-spiking neurons, we had enough neurons in each age group to maintain significant power. Since non-spiking neurons were included, only the current-clamp features that were not related to spiking were included (similar to *Figure 5B*). As illustrated by performance of the prediction models in *Figure 5C and D*, modiolar to pillar gradients in biophysical properties are present even within a narrow age range, confirming that the gradients are not sampling errors resulting from pooling the data over wide age ranges. This is reassuring since the biophysical properties of SGN change with maturation (as we show in the next section) and our dataset is sparse.

Consistent with the overall data set, even in this narrow age group, we found significant collinearity between current-threshold and response latency (r(27) = −0.59, p=0.0008). Using response

**Table 3.** Multiple variable regression models.

| Figure | Variable | ß | SE | Df | /t/ | VIF | RMSE | p-value |
|---|---|---|---|---|---|---|---|---|
| 5A | $I_{threshold}$ | 0.0099 | 0.0016 | 11 | 6.102 | 2.420 | 0.090 | 0.0004 |
| | $g_{max}$ | −0.0182 | 0.0047 | | 3.858 | 3.858 | | |
| | $\tau_{inact}$ | 0.0012 | 0.0005 | | 2.625 | 2.165 | | |
| 5B | Latency | −0.0231 | 0.0045 | 24 | 5.158 | 1.241 | 0.136 | <0.0001 |
| | $\tau_{inact}$ | 0.0005 | 0.0004 | | 1.404 | 1.241 | | |
| 5C | Latency | −0.0192 | 0.00454 | 14 | 4.23 | 1.00 | 0.123 | 0.0010 |
| 5D | Latency | −0.0310 | 0.0067 | 16 | 4.62 | 1.00 | 0.140 | 0.0003 |

latency as the sole input variable for the model at P3-P5 accounts for 55% of the variance in the data ($R^2 = 0.55$, p=0.0010) and predicts the basal position within a 12.3% margin of error (RMSE = 0.123) (*Figure 5C*).

At P6-P8, response latency, resting potentials, $g_{-30}$, and $g_{max}$ were all significantly correlated with the basal position (*Table 2*). Response latency still accounted for 55% of the variance ($R^2 = 0.55$, p=0.0003) and predicted the basal position within a 14% margin of error (RMSE = 0.140) (*Figure 5D*). These narrow-age-binned models show that significant spatial gradients in biophysical properties exist in early post-natal days and are maintained until the onset of hearing.

In summary, we employed current- and voltage-clamp features in different combinations within multiple-linear-regression models to describe the systematic variation of SGN electrophysiology as a function of terminal contact position on inner hair cells. The common underlying factor relating current threshold and latency appears to be the net input conductance of the neurons (as illustrated in *Figure 5E and F*), as probed here by $g_{-30}$ in voltage clamp. As net conductance increased, larger currents were needed to depolarize the SGN somata to threshold, as would be consistent with Ohm's Law (*Figure 5E*). In a similar broad view, larger conductance would also likely translate into faster membrane time constants (as described by time constant $RC = \frac{C}{g}$). Faster membrane time constants are likely to yield shorter first spike latencies. In *Figure 5F*, we plot response latency for all the labeled and unlabeled neurons from the present study against $g_{-30}$. Consistent with the above suggestion, response latency is inversely related to net conductance, as expected if latency is proportional to membrane time constant. These results suggest that spatial gradients in SGN biophysical properties that predict normalized basal position have an underlying dependence on the net conductance seen by the recording electrode.

## Spatial and developmental gradients

Next we examined age-dependent trends in SGN biophysics (*Figure 6*). The three key predictors for NBP change significantly with age; $g_{max}$ becomes larger (r(136) = 0.50, p<0.0001), current thresholds increase (r(79)=0.42, p<0.0001), and average latencies become shorter (r(133)=-0.40, p<0.0001). Note that all recordings between P1 and P16 are included in this analysis of age dependence, even those SGN that did not label to the terminal. To illustrate the range of responses at any particular age range and age dependent changes in biophysics, in *Figure 6A* we plot the size of the near steady-state outward current (at 400 ms) for individual cells as a function of the command voltage. The data are binned into three age groups (P1-5, n = 67; P6-9, n = 57; and P10-16, n = 13). Although the average current is smaller at younger ages, the overall range in current amplitudes is large, even within any particular age group. For example, many SGN at older ages have currents as small as those at younger ages.

### Persistence of biophysical heterogeneity

The degree of biophysical heterogeneity within an age group is illustrated by plotting the distributions for $g_{max}$, current threshold and latency in the same three age bins (*Figure 6B,D and F*). Here, we show the data with color coding to indicate unlabeled neurons (gray), Type II neurons (black) and Type I neurons (from red to blue to indicate normalized basal position from the modiolar to pillar face). The mean and standard deviation for $g_{max}$ and current threshold increases with age (*Figure 6B and D*). In contrast, the mean and standard deviation for latency decreases with age (*Figure 6F*). To compare the relative dispersion of these distributions given the changes in mean value, we computed the coefficient of variation (CV) by taking the standard deviation of the distributions divided by the mean. CV did not change systematically with age for any of the three variables (CV indicated above each distribution in *Figure 6B,D and F*); meaning that the biophysical properties are maintaining a similar degree of heterogeneity across the three age groups. These data show that although the biophysical properties of SGN continue to change with age, they remain biophysically heterogeneous.

### Spatial gradients in biophysical maturation

As we already described, the biophysical properties of SGN are found in a gradient based upon the normalized basal position. These gradients are evident when the distribution of biophysical properties is displayed binned into three age groups (*Figure 6B,D and F*; note the gradation in color from

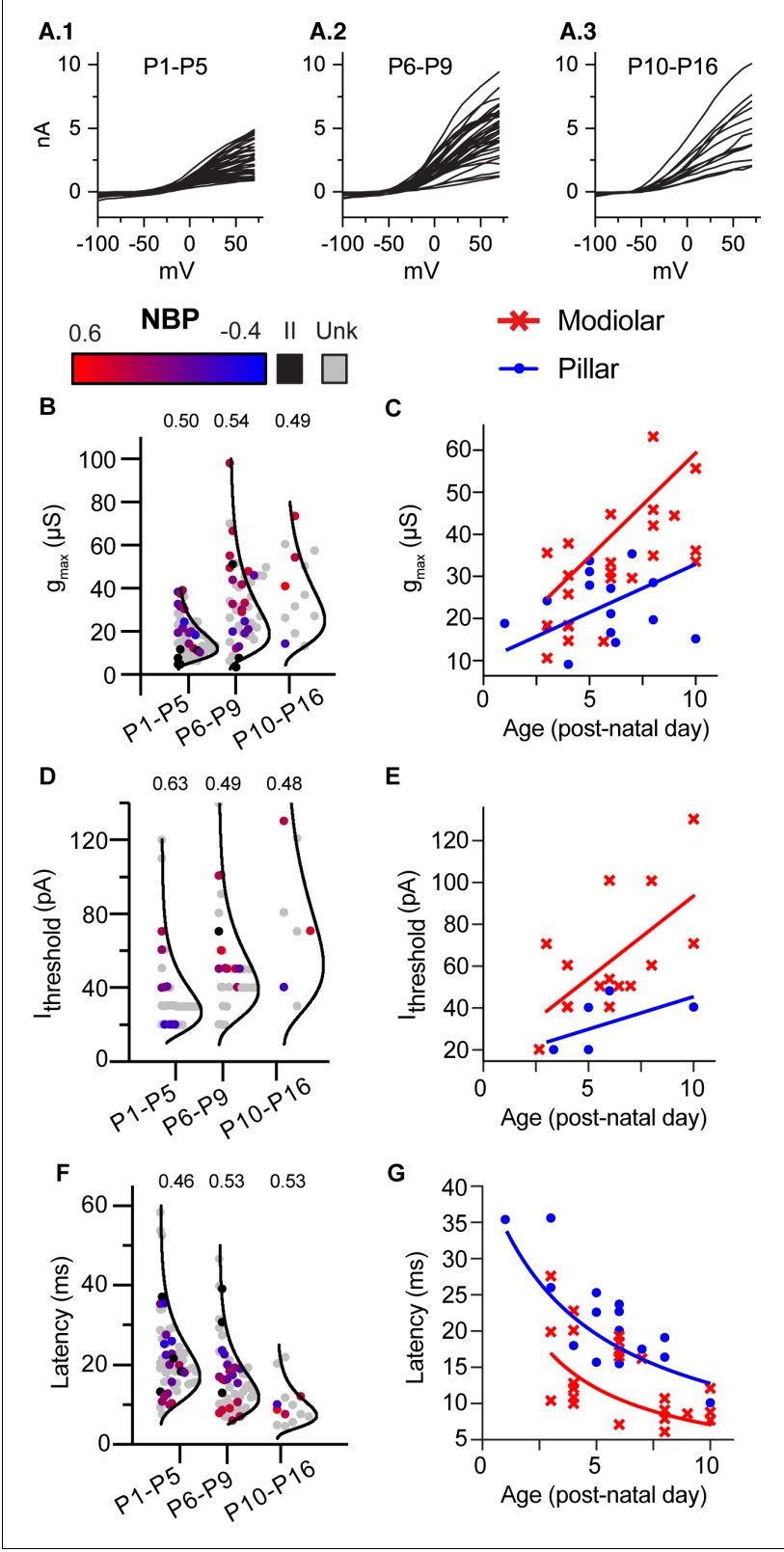

**Figure 6.** Biophysical properties of SGN change with age but remain diverse throughout 3 weeks of post-natal development. (**A**) 1-A.3 Steady-state current/voltage curves show that net outward currents grow in three age bins (P1-P5; P6-9; and P10-P16). Small and large currents are evident in all age ranges. (**B,D and F**) Log-normal plots showing the distribution of maximum conductance ($g_{max}$), current threshold ($I_{threshold}$) and latency in three age

*Figure 6 continued on next page*

*Figure 6 continued*

bins, respectively. Points are color coded to indicate normalized basal position (red to blue gradient), unlabeled cells (gray) and type II neurons (black). Within each age bin, biophysical properties of type I SGN change as a gradient along the NBP scale. The relative dispersion in each distribution is quantified by the CV (coefficient of variation indicated above each distribution) and is not systematically dependent on age. **C,E and F** Plots of same three features, $g_{max}$, current threshold, and latency plotted with age as a continuous variable but with spatial position binned as 'modiolar-contacting' (NBP >0; red symbols) and 'pillar-contacting' (NBP <= 0, blue symbols). Solid lines in **C** and **E** show the best regression fits through modiolar-contacting and pillar-contacting groups ($g_{max}^{mod} = 4.93 * Age + 10; g_{max}^{pillar} = 2.29 * Age + 10$). Solid lines in **F** plot curves defined by the inverting the regression plots in C; $L_{(mod,pil)} = 420/g_{max}^{(mod,pil)}$.

The online version of this article includes the following source data for figure 6:

**Source data 1.** Maximum conductance versus age for *Figure 6C*.
**Source data 2.** Current threshold versus age dataset for *Figure 6E*.
**Source data 3.** Latency versus age dataset for *Figure 6G*.

red to blue indicating the normalized basal position). Indeed, the presence of biophysical gradients at early and late post-natal ages accounts for the success of the age-filtered prediction models presented earlier (*Figure 5C and D*).

Given that the biophysical properties of the entire SGN are changing with age, we examined the spatio-temporal gradients in SGN biophysics. To do this, we performed an analysis of covariance for each biophysical property; with age and position as covariates. The analysis was limited to Type I neurons in which we had a clearly identified normalized basal position. Moreover, we binned normalized basal position into either modiolar-contacting or pillar-contacting groups (*Figures 6C, E and G*). We defined the modiolar-contacting group as SGN with NBP values greater than or equal to 0 (red symbols) and the pillar-contacting group as SGN with NBP values less than 0 (blue symbols). As in the overall dataset; all three variables: $g_{max}$, current threshold and latency were significantly dependent on age and position (*Figure 6C, E and G*; *Table 1*). There was also a significant interaction between the effects of age and synaptic position (modiolar/pillar) on $g_{max}$ (Class*Age: F(1,33) =5.98, p=0.0199, ANCOVA; *Table 1*, *Table 4*). As evident in *Figure 6C*, $g_{max}$ is on average larger *and* grows nearly twice as fast with age for modiolar-contacting SGN than for pillar-contacting SGN. This is indicated by the steeper slope of the regression line for the modiolar-contacting group (red dashed line) compared to that for the pillar-contacting group (blue line).

Current thresholds are also significantly dependent upon age and synaptic position (*Table 1*). Consistent with expectations based on Ohm's law, current thresholds grow with post-natal age and are larger for the modiolar-contacting group. The difference in slope between the modiolar- and pillar-contacting regressions does not reach significance but this could be because sample sizes are smaller for current threshold (Class*Age; F(1,17)=0.64, p=0.43, ANCOVA, *Table 1*, *Table 4*).

Although $g_{max}$ and current thresholds increase with age, the opposite trend is seen for latency (*Figure 6G*). This is because latency is inversely related to conductance (*Figure 5*). Thus, as $g_{max}$ increases with age, latency decreases. In *Figure 6G*, we plot latency as a function of age. Again, latency is significantly dependent on both age and spatial position with the modiolar-contacting group having shorter latencies (*Table 1*). The shorter latencies for the modiolar-contacting group is consistent with their having larger conductances. As with $g_{max}$, there is also a significant interaction between the effects of age and synaptic position on latency (F(1,33)=8.4, p=0.0066, ANCOVA, *Table 1*, *Table 4*). The difference in the rate at which latency changes with age for the modiolar-contacting and pillar-contacting groups can be understood as arising from the difference in the rate at which conductance is changing with age for the two groups. The red and blue lines overlaid on the data points (*Figure 6H*) are two curves defined by the functions,

$$L_M = \frac{K}{g_{max}^M} \tag{2}$$

$$L_P = \frac{K}{g_{max}^P} \tag{3}$$

**Table 4.** Table of regression slopes.

| Figure | Term | Estimate | Standard error | T | p-value |
|---|---|---|---|---|---|
| 3D *Latency* | Slope | −22.8 | 4.02 | −5.66 | <0.0001 |
| | Spike | −2.98 | 8.05 | −0.37 | 0.71 |
| | Non-Spike | 2.98 | 8.05 | 0.37 | 0.71 |
| 4F $g_{max}$ | Slope | 27.87 | 2.55 | 10.89 | <0.0001 |
| | Spike | −0.29 | 15.4 | −0.02 | 0.98 |
| | Non-Spike | 0.29 | 15.4 | 0.02 | 0.98 |
| 4 G$_{g-30}$ | Slope | 17.46 | 0.80 | 10.6 | <0.0001 |
| | Spike | 5.1 | 6.7 | 0.76 | 0.45 |
| | Non-Spike | −5.1 | 6.7 | −0.76 | 0.45 |
| 4 H$_{V1/2}$ | Slope | −19.98 | 9.8 | −2.04 | 0.0495 |
| | Spike | −15.22 | 19.62 | −0.78 | 0.44 |
| | Non-Spike | 15.22 | 19.62 | 0.78 | 0.44 |
| 4I $\tau_{inact}$ | Slope | 178.9 | 62.3 | 2.87 | 0.0084 |
| | Spike | −5.46 | 124.63 | −0.04 | 0.97 |
| | Non-Spike | 5.46 | 124.63 | 0.04 | 0.97 |
| 6C $g_{max}$ | Slope | 2.80 | 1.06 | 2.63 | 0.0127 |
| | Pillar | −2.60 | 1.06 | −2.45 | 0.0199 |
| | Modiolar | 2.60 | 1.06 | 2.45 | 0.0199 |
| 6E $I_{threshold}$ | Slope | 4.68 | 2.26 | 2.07 | 0.054 |
| | Pillar | −1.81 | 2.26 | −0.8 | 0.4334 |
| | Modiolar | 1.81 | 2.26 | 0.8 | 0.4334 |
| 6G *Latency* | Slope | −1.752 | 0.29 | −5.97 | <0.0001 |
| | Pillar | −0.85 | 0.29 | −2.9 | 0.0066 |
| | Modiolar | 0.85 | 0.29 | 2.9 | 0.0066 |

where *K* is a proportionality constant and $g^{M}_{max}$ and $g^{P}_{max}$ are the linear fits (from *Figure 6G*) that best describe the age dependence of $g_{max}$ on the modiolar and pillar faces, respectively. The fit for the pillar face was constrained to intersect that of the modiolar face between P0 and P3. The curves defined by *Equations 5 and 6* are successful at describing the difference in latencies between the two groups; pillar-contacting fibers have longer latencies than do the modiolar-contacting fibers and modiolar-contacting fibers are developing faster than are pillar-contacting fibers. By P10, $L_{P}$ is about 13 ms, making the fibers contacting the pillar face about 6 days delayed in maturation on average compared to those on the modiolar face. The spatial and maturational gradients are qualitatively similar, raising the possibility that diversity in SGN biophysical properties reflects a local spatial gradient in maturation.

## Pruning of terminal branches is faster in modiolar-contacting SGN

Next, we describe fiber morphology as supporting evidence for modiolar-contacting neurons developing at a faster rate than pillar-contacting neurons. Our results suggest that morphological

maturation parallels biophysical maturation. The afferent terminals of SGN are highly branched during early development and are pruned during maturation to form a single bouton connection between the SGN and inner hair cell (*Druckenbrod and Goodrich, 2015*, *Huang et al., 2007*, *Echteler, 1992*). Consistent with previous reports, we also found that the number of terminal branches decreases as a function of maturation (r(53) = −0.79, p<0.0001) (*Figure 7A and B*). We observed mature one-to-one connections as early as P5 on the modiolar face. In contrast, none of the pillar-contacting fibers had mature terminal morphology, even by P10 (*Figure 7C*). We quantified differences in the pruning rate of modiolar- and pillar-contacting fibers from P5-P10 and found that modiolar-contacting fibers' have fewer branches than do pillar-contacting fibers (1.9 +/- 0.24, n = 12 vs. 3.2 +/- 0.32, n = 7, respectively) (p=0.030, t-test) (*Figure 7C*, inset). Since, modiolar-contacting fibers appear to prune at a faster rate than pillar-contacting fibers, these data are consistent with the biophysics in suggesting that neurons on the modiolar face are changing faster with post-natal than those on the pillar face.

In mature auditory nerve, fiber diameters are larger for high-SR fibers, which are the fibers primarily found on the pillar side of the inner hair cell (*Merchan-Perez and Liberman, 1996*). We measured fiber diameters of fluorescently labeled SGN as the average diameter of the last 10–20 µm of the parent fiber just before the branching begins near the inner hair cells (*Figure 7A*, see bracketed

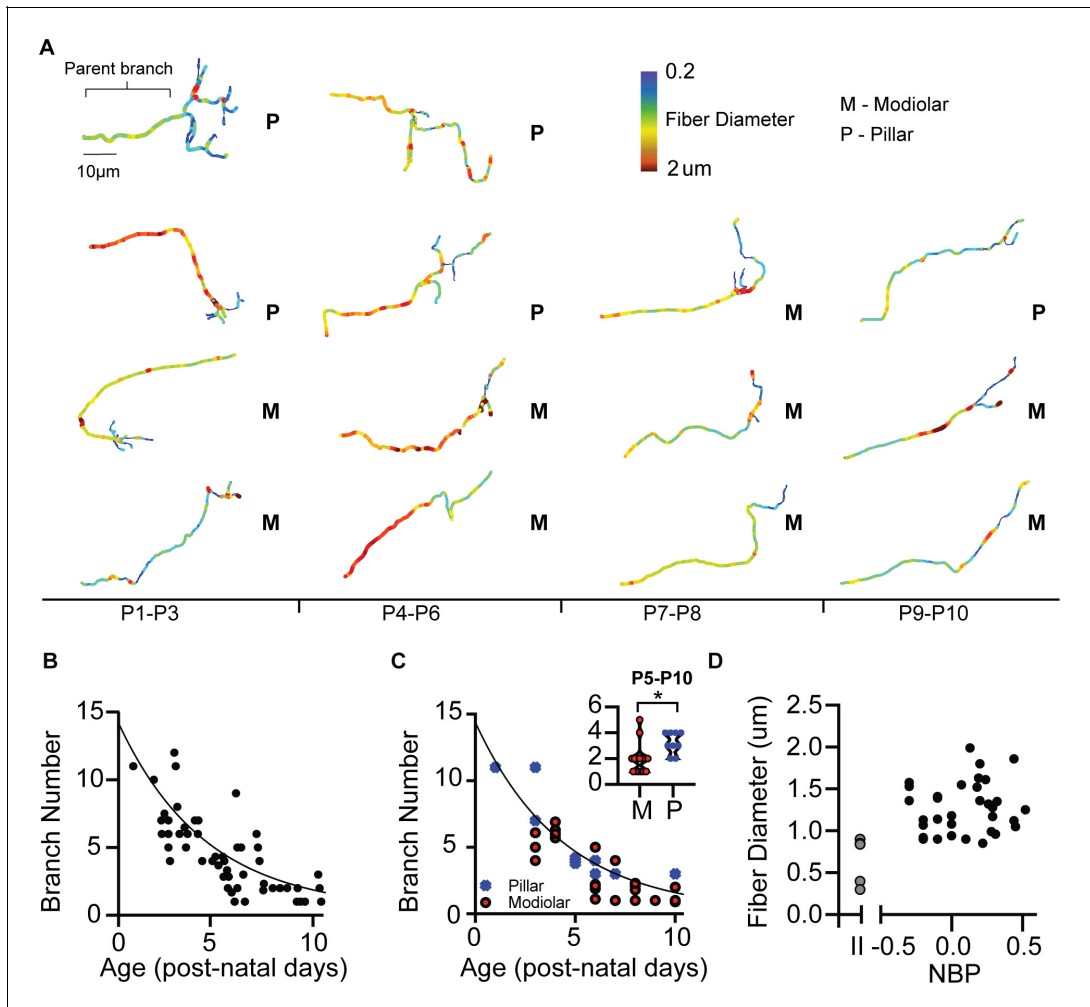

**Figure 7.** Morphological development of spiral ganglion terminal branches. (**A**) Three-dimensional reconstructions of SGN terminals are made to quantify the number of branches at each terminal and the approximate diameter of the fibers. (**B**) The number of branches exponentially decreases over the course of development. One-to-one connections with the inner hair cell can be seen at P5. In some cases, terminals have multiple branches at P10. (**C**) Pillar-contacting SGN are not observed to have the mature one-connection morphology. (**D**) Fiber diameters do not differ between type I SGN along the NBP scale. Type II SGN have significantly smaller fiber diameters than type I SGN.

area). Consistent with previous in vivo labeling studies, we found that type I SGN have significantly larger fiber diameters than do type II SGN (1.28 +/- 0.05 µm, n = 37 vs. 0.59 +/- 0.12 µm, n = 6; p<0.0001, t-test) (*Figure 7D*). Fiber diameters are not significantly different between modiolar-contacting SGN and pillar-contacting SGN (1.25 +/- 0.29 µm vs. 1.11 +/- 0.22 µm; p=0.25, t-test) (*Figure 7D*). These results are consistent with reports in mice where fiber diameters begin to differentiate around the onset of hearing (Coates, personal communication).

## Discussion

### Biophysical properties of SGN vary as spatio-temporal gradients along the base of the inner hair cell

Our results show striking correlations between the biophysical properties of SGN somata and the position where peripheral dendrites contact hair cells. Within the type I population, current thresholds decreased, first-spike latencies increased, and the size of net outward conductance decreased in a gradient as as a function of contact position along the base of inner hair cells. Although the ion channel properties of SGN somata are known to be diverse in vitro, this is the first clear demonstration that such diversity is systematically organized about the auditory nerve's effective map for 'intensity-coding'. Specifically, the more excitable neurons with low current thresholds and long membrane time constants in vitro are found on the pillar side of inner hair cells, where in vivo auditory nerve fibers have high spontaneous rates and low-intensity thresholds (*Kiang et al., 1965*; *Liberman, 1978*). The direction of in vitro biophysical gradients qualitatively aligns with mature in vivo physiology and hints at a possible post-synaptic contribution to sound-driven thresholds and excitability.

### SGN diversification during maturation

Biophysical diversity is not a transient feature of immaturity and likely persists to influence mature function. Although SGN biophysics continue to change during development, we found that neurons do not become more similar as they mature. The neurons contacting the modiolar face of the hair cell change faster with age than do the pillar contacting neurons. As a result, SGN display a wide range of biophysical properties at both early and late post-natal ages. The coefficient of variation (a measure of relative dispersion) for the distributions of biophysical features does not change with maturation. Consistent with this result, single-cell RNA sequencing recently demonstrated that the molecular and transcriptomic profile of SGN changes during development to eventually stabilize into distinct subtypes (*Petitpré et al., 2018*; *Shrestha et al., 2018*; *Sun et al., 2018*).

The biophysical gradients we describe at the local scale under an inner hair cell are reminiscent of previously described global (i.e. along the length of the cochlea) gradients in SGN biophysics. By the end of the second post-natal week, firing patterns are more rapidly accommodating and first-spike latencies are on average faster than in the first week; more so in fibers contacting the high-frequency-encoding basal turn of the cochlea than in the low-frequency-encoding apical turn (*Crozier and Davis, 2014*; *Adamson et al., 2002*). Across the entire spiral ganglia, firing patterns become more rapidly accommodating and first-spike latencies become faster as the animal matures (*Crozier and Davis, 2014*; *Adamson et al., 2002*). These global spatio-temporal biophysical gradients likely reflect a global cochlear base-to-apex maturational gradient similar to that previously described in cochlear hair cells (*Waguespack et al., 2007*; *Lelli et al., 2009*).

Our data from the middle-turn of the cochlea suggest that the global pattern of spatio-temporal development is being recapitulated on a local scale under individual inner hair cells. Here, we showed that the biophysical properties of SGN contacting the modiolar face change faster with age, acquiring large conductances and faster first spike latencies, than that of SGN contacting the pillar face. The morphology of SGN dendrites suggests that this may reflect a local gradient in maturation. The pruning of dendritic branches to form a one-terminal-contact is a well-described morphological hallmark of maturing auditory neurons (*Echteler, 1992*; *Barclay et al., 2011*; *Huang et al., 2007*). We found that the terminal branches of many modiolar-contacting SGN fibers are pruned by an earlier age than those of pillar-contacting fibers. Evidence from the literature further supports this suggestion; for example, modiolar-contacting fibers appear to be less enriched for the calcium-binding protein, calretinin, than pillar-contacting fibers in mature rats (*Kalluri and Monges-Hernandez,*

*2017*) and mice (e.g. *Shrestha et al., 2018*). This difference in protein expression (among others) emerges only after a period of maturation (*Shrestha et al., 2018*; *Sun et al., 2018*; *Petitpré et al., 2018*). Perhaps, the expression of calretinin in pillar-contacting fibers is also a mark of residual immaturity. That pillar-contacting fibers may be developing at a slower rate is consistent with the observation that high-spontaneous-rate/low-threshold responses (prevalent for fibers on the pillar face) emerge later in development than do low-spontaneous-rate/high-threshold responses (*Wu et al., 2016*). Taken together with the literature, our data suggest that local gradients in the biophysical properties of SGN are supported by local gradients in maturation.

## Factors contributing to biophysical diversification

The biophysical properties of SGN appear to depend on extrinsic factors, at least in part. In cultured SGN, the presence of growth factors such as BDNF and NT-3 (typically released by hair cells and supporting cells in situ) has a profound influence on firing patterns and expression of proteins for ion channels and receptors (e.g. *Flores-Otero et al., 2007*; *Sun and Salvi, 2009*). The influence of these growth factors also follows a global cochlear-base to cochlear-apex gradient; with higher levels of BDNF promoting rapidly accommodating firing patterns in the base of the neonatal cochlea. In contrast, NT-3 is found at higher levels in the apex of the cochlea where firing patterns are more slowly accommodating. That extrinsic signals may influence SGN diversity is further strengthened by recent findings that post-natal diversification fails in the absence of synaptic input (*Shrestha et al., 2018*; *Sun et al., 2018*).

Although the molecular diversity of SGN appears to be partly dependent on input from the hair cell, the interactions may be reciprocal. In zebrafish, the presence of afferent fibers is crucial for the formation and stabilization of synaptic ribbons in the hair cell (*Suli et al., 2016*). In mice, *Sherrill et al., 2019* recently reported that spatial gradients in the voltage dependence of pre-synaptic calcium channels is controlled by the enhanced expression of the Pou4f1 transcription factor in modiolar-contacting SGN. This would mean that post-synaptic heterogeneity may be important for driving pre-synaptic heterogeneity. These results suggest that functional heterogeneity of auditory nerve responses may be shaped by complex reciprocal signaling between hair cells and spiral ganglion neurons during post-natal development.

## Comparison between acute and cultured preparations

SGN electrophysiology in the acute semi-intact preparation is similarly diverse to that reported in cultured ganglionic preparations. We found that SGN in the acute preparation were diverse in action potential properties and size of whole-cell conductance. SGN with large conductances tended to be rapidly accommodating, produced action potentials with fast latencies and in response to large current injections. These neurons tended to contact the modiolar-face of the hair cell. These findings are consistent with SGN biophysics in cultured neurons, where rapidly accommodating neurons have faster latencies, higher current thresholds and larger low-voltage gated potassium currents (*Mo and Davis, 1997*; *Chen and Davis, 2006*; *Liu and Davis, 2014b*; *Lv et al., 2010*).

Our results are also consistent with the idea that modiolar-contacting SGN may have larger concentrations of low-voltage-gated potassium conductances ($g_{KL}$) than pillar-contacting neurons. This suggestion is based on our observation that the neurons have larger whole-cell conductance values at negative potentials (i.e. $g_{-30}$ values). Although we did not isolate specific ion channel currents (e.g. low-voltage gated currents conducted by channels such as Kv1 and KCNQ), our results are consistent with previous studies in which threshold and spike-train accommodation of SGN firing patterns were controlled by these channels (*Liu et al., 2014a*; *Lv et al., 2010*). Recent single-cell RNA-sequencing reports are consistent with our results in showing that enrichment for potassium channels (including high-voltage-gated potassium channels (Kv4)) is greater in modiolar-contacting SGN (*Shrestha et al., 2018*; *Sun et al., 2018*; *Petitpré et al., 2018*).

The inclusion of the graded-response group and low-incidence of the slowly accommodating group may be a difference between the present study and some previous in vitro studies of SGN. We do not believe that graded responses are coming from damaged cells because the recordings were made after forming giga-Ohm seals, had stable resting potentials, made for clean biocytin labeling. In our experience, biocytin labeling was not successful in damaged cells. As our results show, key membrane properties such as response latencies were similar in both graded and spiking

neurons, allowing us to pool these cells into a unifying analysis group. Graded responses have also been observed in two other studies where SGN were prepared without a significant period in culture (*Jagger and Housley, 2003*; *Santos-Sacchi, 1993*). One possible explanation is that culturing may redistribute ion channels to portions of the membrane that are otherwise covered by myelin in the endogenous condition (as described in CNS by *Shrager, 1993*).

Another potential difference between this and previous in vitro studies stems from our recording in semi-intact preparations where the presence of peripheral dendrites could make recordings non-isopotential. We excluded the possibility that non-isopotential behavior confounds the data in finding that the capacitive currents in response to small voltage steps were decaying with single-exponential time courses. This suggested to us that we were looking at largely isopotential compartments mostly likely confined to SGN somata (see supplement). Whether developmental changes in myelination influence this interpretation will require additional work characterizing the formation of myelin around the somata. Thus, aside from differences that might arise from culturing, we believe that the interpretation of ion channel properties is similar in this semi-intact preparation as for recordings from isolated ganglionic preparations.

## The biophysical properties of type II SGN compared to type I SGN

Type II SGN represent ~5% of the SGN population, make connections with multiple outer hair cells, and have a yet unknown role in the neuronal encoding of sound (*Spoendlin, 1969*; *Kiang et al., 1984*; *Liberman and Dodds, 1984*). In this study, type II SGN were 9 of 59 labeled fibers ranging in age from P3 to P8. The small number of neurons over a wide age range limits our ability to analyze and attribute a statistically significant biophysical phenotype to type II. Also, there may be more than one subtype of type II fibers (*Vyas et al., 2019*), making the scarcity of type II's even more limiting for statistical evaluation. However, we found two biophysical trends that distinguish type II SGN from the majority of type I SGN. Type II fibers tended to be on the extreme end of the distribution in having small steady-state outward potassium currents that inactivated rapidly (*Figure 4*). The values of both of these voltage-clamp features have a large variance, with type II SGN positioned at the extrema of the two distributions. That the currents of type II fibers inactivate faster than for type I fibers is consistent with the previous reports of *Jagger and Housley, 2003*. Our results extend those previous results by showing that the inactivation time constant of type I SGN varies as a gradient from the pillar to modiolar face. Although the voltage-clamp responses showed a trend relative to type II and type I SGN, we did not find significant differences in response latencies in current clamp for the type II SGN. The latencies of type II SGN were not significantly different from the mean latencies of type I SGN. Because of this overlap, we were not able to use these variables to predict differences between type I and type II SGN.

## The relevance of somatic recordings

The spike initiation zone for auditory neurons is on their peripheral neurites under inner hair cells, not at their somata where we are recording. It remains to be tested whether ion channels are found in spatial gradients at the peripheral neurites. Such gradients are possible since similar groups of ion channels and neurotransmitter receptors are seen at the soma and peripheral neurite (*Rutherford et al., 2012*; *Yi et al., 2010*; *Hossain et al., 2005*). Non-uniform sampling due to difficulty in accessing terminals with patch-electrodes may prevent such studies from seeing the full biophysical diversity seen in the SGN somata (*Rutherford et al., 2012*). Thus, whether spontaneous rate and threshold are defined exclusively by pre-synaptic mechanisms or further shaped by post-synaptic mechanisms remains an unresolved issue. Indeed, the presence of lateral olivocochlear efferents on the terminals of auditory neurons has long made the possibility of post-synaptic modulation of spiking activity highly likely (*Wu et al., 2020*).

How would variations in post-synaptic biophysics shape auditory nerve responses? Neurons with differences in current thresholds and temporal integration properties are likely to respond differently to the heterogeneity in synaptic input that inner hair cell synapses are known for. Specifically, excitatory post-synaptic currents (EPSCs) at individual inner hair cell synapses are remarkably heterogeneous in amplitude and kinetics (e.g., *Grant et al., 2010*). Some EPSCs are monophasic (temporally compact EPSCs with fast onset kinetics and typically large amplitudes) while others are multiphasic EPSCs (broad, slow onset EPSCs with smaller amplitudes). One proposal is that the relative

prevalence of monophasic/multiphasic EPSCs at different synapses may contribute to diversity in the responses of auditory afferents (e.g. *Grant et al., 2010*). However, whether the prevalence of mono- and multi-phasic EPSCs varies with synaptic position remains to be determined. Our results suggest an additional level of post-synaptic processing that could further shape the responses of auditory neurons. Pillar-contacting SGN likely have low-current thresholds and longer temporal integration windows. These neurons will be more likely to reach spike threshold in response to a larger variety of EPSC amplitudes and shapes. The lower current threshold means that these neurons will spike in response to small and large EPSCs and the longer temporal integration means that these neurons can also reach threshold by integrating over the slow onset kinetics of multi-phasic events. Since intermediately accommodating firing patterns were more prevalent on the pillar face (*Figure 2*), supra-threshold multiphasic EPSCs may also produce more than one action potential. Ectopic spiking has been noted in pre-hearing auditory afferents (*Wu et al., 2016*). In contrast, fibers with larger current thresholds and shorter temporal integration times are more likely to respond to large-amplitude, rapid onset EPSCs. Such fibers would presumably also experience more spike failure in response to sub-threshold events and the slower depolarizations that would result from multiphasic EPSCs. Whether such spike failure occurs in this system remains unclear given the difficulty of terminal recordings; some studies report that each EPSC faithfully produces an action potential (e.g., *Rutherford et al., 2012*), while a recent study reported that in some fibers many EPSCs do not produce action potentials (e.g. *Wu et al., 2016*). Our results suggest that post-synaptic heterogeneity could dove-tail with pre-synaptic heterogeneity to shape diversity in auditory neuron responses.

Somatic ion channels may also have a function in and of themselves. According to models, the expression of ion channels on the SGN soma may be helpful for preventing spike failure. The impedance mismatch between the peripheral dendrite and the large soma can dramatically slow down and attenuate spikes, thereby increasing the probability of spike failure (e.g. modeling by *Rattay et al., 2013*; *Hossain et al., 2005*). In this scenario, the soma could independently filter spike trains as they sweep towards the brainstem (*Davis and Crozier, 2015*), and the nature of that filtering would be determined by somatic ion channels.

The observed link between in vitro cellular biophysics with putative SR-subgroups opens new opportunities for exploring and reevaluating mechanisms driving SGN function, regardless of the ultimate functional consequence of gradients in somatic ion channel properties. We have described a prediction model which uses easily measured biophysical properties such as current threshold and first-spike latency to classify SGN into putative SR-subgroups. Moreover, the modeling strategy systematically identified covariant/redundant variables among a larger group of easily measured biophysical variables. This identified gradients in potassium conductance as the most parsimonious explanation for the group of biophysical gradients observed (*Figure 5*). We argue that systematic variation in net-conductance accounts for the gradients in current threshold and response latencies reported here. As a result, highly covariant variables such current threshold and latency can interchangeably predict basal position. By combining non-redundant biophysical variables (such as current threshold and outward current inactivation), our regression models predict basal position to within a +/- 9% margin of error. This is a remarkable resolution given that the inputs to the model use relatively simple biophysical measurements. The success of the model is a reflection of the strength of these underlying relationships.

A final practical outcome of these results is that we can apply the prediction model to study other properties of SR-subgroups by using technically favorable dissociated somatic preparations. These would include studies focused on understanding the specific ion channels expressed by different SGN subgroups, the sensitivity of these neurons to different efferent neurotransmitters and the vulnerability of SGN subgroups to degeneration (*Furman et al., 2013*). This work complements the search for molecular markers for SGN subgroups because these approaches are feasible in neonatal animals before molecular expression has matured and in non-mouse species where transgenic models are not readily available.

# Materials and methods

**Key resources table**

| Reagent type (species) or resource | Designation | Source or reference | Identifiers | Additional information |
|---|---|---|---|---|
| Strain, strain background (species) | Long-Evans (rat) | USC Vivarium and Charles River | RRID:RGD_2308852 | Freshly isolated cochlear middle turns |
| Antibody | anti-myosin-VI (rabbit polyclonal) | Proteus Biosciences | Cat# 25–6791, RRID:AB_10013626 | IF(1:1000) |
| Antibody | anti-peripherin (rabbit polyclonal) | Millipore | Cat# 183 AB1530, RRID:AB_90725 | IF(1:500) |
| Antibody | streptavidin Alexa Fluor 488 conjugate | Molecular Probes | Cat# S32354, 182 RRID:AB_2315383 | IF(1:200) |
| Antibody | Alexa Fluor 594 (goat anti-rabbit, polyclonal) | Thermo Fisher Scientific | Cat# A-11080, 185 RRID:AB_2534124 | IF(1:200) |
| Other | Vecta-shield | Vector Laboratories | Cat# H-191 1000, RRID:AB_2336789 | |
| Software, algorithm | JMP | SAS Institute | RRID:SCR_002865 | |
| Software, algorithm | Matlab | Mathworks | RRID:SCR_001622 | |
| Software, algorithm | pClamp | Molecular Devices | RRID:SCR_011323 | |
| Software, algorithm | OriginPro | Origin Labs | RRID:SCR_015636 | |
| Software, algorithm | Imaris | Oxford Instruments | RRID:SCR_007370 | |
| Software, algorithm | Prism 8 | Graph Pad | RRID:SCR_002798 | |
| Software, algorithm | FilamentTracer in Imaris | Oxford Instruments | RRID:SCR_007366 | |

## Preparation

Data were collected from spiral ganglion somata in semi-intact cochlear preparations from Long-Evans rats on post-natal day (P)one through P16. Animals were handled in accordance to the National Institutes of Health Guide for the Care and Use of Laboratory Animals and all procedures were approved by the animal care committee at the University of Southern California.

Temporal bones were dissected in chilled and oxygenated Liebovitz-15 (L-15) medium supplemented with 10 mM HEPES (L-15; pH 7.4,~315 mmol/kg). The semi-circular canals were removed, as was the bone surrounding the cochlea's sensory epithelium. The cochlea was then cut into three turns, with the middle turn used for all experiments. The middle turn was further prepared by removing the Reissner's membrane, stria vascularis, and tectorial membrane. The cochlear turn was then mounted under two stretched nylon threads on a glass coverslip. The entire coverslip with cochlea so mounted was incubated in an enzyme cocktail containing L-15, 0.05% collagenase, and 0.25% trypsin for 15–20 min at 37°C. The digested tissue was washed with fresh L-15 and mounted under the microscope objective. The preparation was then continually perfused with fresh oxygenated L-15.

Somatic visibility was best in the layers closest to the objective. As a result, mounting direction was sometimes flipped (with hair cells oriented to the coverslip) to target neurons in other areas of the ganglion. Angle of electrode approach was also varied to target SGN that were close to the modiolar core and those closer to the organ of Corti.

## Analysis of electrophysiology

All data were analyzed with pClamp 10.7 software. In current-clamp mode, we measured various whole-cell properties of the neuron including the resting potential, current threshold for spikes, action potential latency, and the after-hyperpolarization time constant. In voltage-clamp mode, we measured the approximate steady-state value of outward currents (~taken 400 ms after stimulus onset) in response to a family of voltage steps. In many cells, the outward current inactivated for long-duration voltage step. We quantified the time-course for inactivation, $\tau_{inact}$, by fitting a single exponential curve from the peak outward current to end of the 400 ms. This was computed in all

cells for the largest command voltage applied (+ 70 mV) (*Figure 4C*, inset). In some neurons (n = 39 of 128), a single exponential curve did not provide a good fit or the 400 ms protocol was too short to adequately estimate the time course; these cells were not included in subsequent multiple variable regression analyses.

Cell capacitance (7.4 +/- 0.36 pF) and series resistance (13.9 +/- 1.85 MOhms) were estimated using the membrane-test protocol on pClamp and/or by analyzing a recording of the capacitive transient in response to small hyperpolarizing voltage steps. No online series resistance correction was applied during recordings. All whole-cell currents and stimulus voltages in the main text and figures are reported without correcting for series resistance and without normalizing by cell capacitance. As we discuss in the supplement section, these corrections do not change our interpretations.

## Electrophysiology

Recordings were made between one and five hours after dissection. Preparations were viewed at X630 using a Zeiss Axio-Examiner D1 microscope fitted with Zeiss W Plan-Aprochromat optics. Signals were driven, recorded, and amplified by a Multiclamp 700B amplifier, Digidata 1440 board and pClamp 10.7 software.

Recording and cleaning pipettes were fabricated using filamented borosilicate glass. Pipettes were fired polished to yield an access resistance between 5 and 7 MΩ. The tip of each recording pipette was covered in a layer of parafilm to reduce pipette capacitance. Large bore cleaning pipettes were used to mechanically remove excess tissue debris surrounding the somatic area. A second cleaning pipette was used to apply suction to remove myelinating glial cells and layers of myelin from the spiral ganglion neuron.

Recording pipettes were filled with the following standard internal solution (in mM): 135 KCl, 3.5 MgCl$_2$, 3 Na$_2$ATP, 5 HEPES, 5 EGTA, 0.1 CaCl$_2$, 0.1 Li-GTP, and titrated with KOH to a pH of 7.35. This yielded a total potassium concentration of 165 mM with a total osmolality of 300 mmol/kg. Voltages are reported without correcting a junction potential of approximately 3.8 mV (calculated by JPCalc as implemented in pClamp 10.7, Barry 1994).

In this study, we only used recordings in which the neuron formed a giga-ohm seal and presented a stable resting potential. After recording whole-cell currents and voltage responses in current clamp, we stimulated the cell with current pulse trains which we empirically determined helped drive the biocytin toward the peripheral terminal. Neuronal terminals were successfully filled in ~50% of recordings. Some neurons did not fill if the soma lysed after the recording pipette was pulled away. We found that pipette resistances of 5–7 MΩ were best for successfully sealing, driving biocytin, and cleanly pulling away from the soma.

## Immunohistochemistry

Following the electrophysiology, the preparation was processed to immunohistochemically label hair cells and to attach a fluorescent label to the biocytin filled peripheral processes. The preparations were fixed in 4% paraformaldehyde for 15 min at room temperature followed by three rounds of washing in 5% phosphate-buffered saline (PBS) for 10 min each on a shaker. The preparations were placed for one hour in a blocking buffer consisting of 16% normal goat serum, 0.3% Triton-X, 450 mM NaCl, and 20 mM phosphate buffer. Next, preparations underwent three rounds of wash in 5% PBS for 5 min each. The preparation was then sequentially incubated in the following primary and secondary antibodies dilutions. Primary antibodies were added and incubated for 12–18 hr at room temperature. Secondary antibodies were added and incubated for 1 hr. Following both primary and secondary incubations, we washed the samples for three 15-min rounds in 1x PBS.

Primary antibodies included (1) anti-myosin-VI-rabbit polyclonal to label the cytoplasm of hair cells, (2) streptavidin Alexa Fluor 488 conjugate to label biocytin filled neurons, and (3) anti-peripherin to label type II spiral ganglion neurons. Secondary antibodies included Alexa Fluor 594 anti-rabbit.

## Imaging

The immunolabeled preparations were mounted under a glass coverslip and onto glass slides with the fade-protectant medium Vecta-shield. Hardened nail polish dots were placed under the coverslip to prevent it from crushing the cochlear sections. We generated z-stack images of each preparation

on one of the following two confocal microscopes: (1) Olympus FV1000 laser scanning confocal microscope and (2) Zeiss LSM 800 with a 10-60x, 1.42 numerical aperture, oil immersion objective, and a threefold zoom (on IHC-SGN synapse). The scanning format was set to collect 1024 × 1024 pixels yielding a sampling of 0.069 µm/pixel in the lateral (XY) dimension. Sections were acquired with z-steps of 0.49 µm with pinholes set at 1 Airy unit. We scanned the different fluorescent signals in separate channels in sequence to ensure optimal imagining of each structure.

## Analysis of morphology

Images were processed using Imaris software in order to analyze the connectivity pattern of the SGN afferent projection and the hair cells. In the XY plane of the z-stack images, SGN were primarily classified based on whether the afferent projection contacted the inner hair cell region or projected past the inner hair cells, turned radially and projected into the outer hair cell area. Three-dimensional reconstructions of the afferent projection were produced in Imaris by using the *FilamentTracer* tool in Imaris. The tool reconstructs the fluorescent signal by tracing the fiber (and branches when present) and fills the area with sequential spheres whose diameters best fit the diameter of the fiber at a specific point. Imaris provides multiple measurements of the fiber including the length, diameter, and volume of the afferent fiber on average and/or at a region of interest. The number of branches were counted manually by counting the number of projections extending from the initial parent fiber. The diameter varied along the length of the extending branch from the soma to hair cell. We measured and reported the average diameter of the last 10–20 µm of the parent fiber just before the branching begins near the inner hair cells. In fibers with multiple branches contacting the hair cell, we took the contact point that was closest to the cuticular plate as the primary contact.

## Statistical analysis

For our statistical analysis, we used a combination of pClamp, Matlab, JMP, Origin Pro, and Prism eight software packages. pClamp software was used to gather and quantify raw data from electrophysiological recordings. Imaris and MATLAB were used to quantify the morphology of the SGN. All variables were confirmed as being normally distributed using a Shapiro-Wilk test for small datasets. Equivalence of variance was tested using Levene's test. To compare the means of two distributions, we used a Student's t-test. For multiple comparisons, we used one-way ANOVA followed by Tukey's HSD post-hoc analysis (for groups with equal variances) or Welch's ANOVA and Games-Howell HSD post-hoc tests (for groups with unequal variances). The relative dispersion in distributions was quantified by the coefficient of variation (CV) which is the ratio between the standard deviation and the mean of the distribution. We used 2-way ANCOVA to compare between groups when one covariate was a continuous variable (e.g. when comparing non-spiking vs spiking groups as a function of normalized basal position or comparing modiolar vs. pillar groups as a function of age). The results of the ANCOVA analyses and slope analysis of regressions are shown in *Table 1* and *Table 4*, and the individual data points with regressions are shown in figures. The strength of correlations between two continuous variables is reported by Pearson's r. We used an alpha-level of 0.05 for all statistical tests. In most cases, the full distribution of data is displayed by graphing individual points.

## Model making

We built models based on multiple-linear-regressions to predict normalized-basal position based on the biophysical properties of type I SGN (see Results). Given their small numbers, type II SGN were excluded from the data set used for modeling. The regressions computed a predicted basal position ($y_i$) based on a linear combination of multiple input variables (i.e. biophysical features from current clamp and voltage clamp):

$$y_i = \beta_o + \beta_1 * x_{1,i} + \beta_2 * x_{2,i} + \beta_3 * x_{3,i} + \ldots + \beta_n * x_{n,i} + \epsilon_i$$
$$y_i = \hat{y}_i + \epsilon_i$$

(4)

The input variables $\left(x_{1,2,3,\ldots,n}\right)$ contributing to the regression were those with the most significant correlations with normalized basal position (*Table 2*). Next, the regression coefficients and intercepts $(\beta_{1,2,3,\ldots,n} \, and \, \beta_o)$ were optimized to minimize the sum of the squared error between the predicted basal position ($y_i$) and the actual basal position ($y_i$):

$$\sum_i \epsilon_i^2 = \sum_i (y_i - \hat{y}_i)^2 \tag{5}$$

To test for redundancy across variables we used *Equation 6* to compute the variable inflation factors (*vif*):

$$vif(\beta_n) = \frac{1}{1 - R_n^2} \tag{6}$$

where $R_n^2$ is the $R^2$ obtained when the $n^{th}$ variable ($x_n$) is regressed against the remaining variables.

Variables with *vifs* greater than 4.0 were deemed to be highly collinear and thus were eliminated as containing redundant information. We used a backward stepping procedure to successively eliminate input variables such that the fewest possible input variables provided the optimal adjusted-$R^2$ and root-mean-squared error (RMSE). This method of variable elimination is an especially aggressive method for minimizing overfitting. Power analysis was performed in JMP for each statistical test to determine observed power and minimum sample size with a significance level (alpha) of 0.05. Power and least significant number were determined by the significance level, error standard deviation (sigma), and effect size (delta). The identity of the variables $x_i$, coefficients $\beta_i$ and *vif* values for four models corresponding to four data subsets are reported in *Table 3*.

## Acknowledgements

We acknowledge the excellent technical support provided by Ms. Maya Monges-Aviles. We thank Ruth Anne Eatock, Christopher Shera, and members of the Kalluri laboratory and the Caruso Department of Otolaryngology for their valuable comments at various stages of this work.

## Additional information

### Funding

| Funder | Grant reference number | Author |
|---|---|---|
| National Institutes of Health | R01DC015512 | Radha Kalluri |
| National Institutes of Health | R03DC012652 | Radha Kalluri |
| American Otological Society | | Alexander L Markowitz Radha Kalluri |

The funders had no role in study design, data collection and interpretation, or the decision to submit the work for publication.

### Author contributions

Alexander L Markowitz, Data curation, Formal analysis, Funding acquisition, Validation, Investigation, Visualization, Writing - original draft; Radha Kalluri, Conceptualization, Resources, Data curation, Software, Formal analysis, Supervision, Funding acquisition, Validation, Investigation, Visualization, Methodology, Project administration, Writing - review and editing

### Author ORCIDs

Radha Kalluri (iD) https://orcid.org/0000-0002-0360-8965

### Ethics

Animal experimentation: Animals were handled in accordance to the National Institutes of Health Guide for the Care and Use of Laboratory Animals and all procedures were approved by the animal care committee at the University of Southern California (protocol number 20704).

### Decision letter and Author response

Decision letter https://doi.org/10.7554/eLife.55378.sa1

Author response https://doi.org/10.7554/eLife.55378.sa2

## Additional files

### Supplementary files
• Transparent reporting form

### Data availability
Source data files are provided as an excel spreadsheet.

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

## Appendix 1

We quantified the net polarization of fibers using a metric we call polarization vector (see Materials and methods). To do this, we assigned a unitary value and sign to each branch depending on polarization (+one for modiolar and –one for pillar). The net polarization of the fiber was computed by taking the sum of the individual polarizations divided by the total number of branches (*Figure 1H*). A polarization vector value of 1 indicates that all of the branches of a fiber are on the modiolar side of the inner hair cell, –one indicates that all branches are on the pillar side, and zero means that the branches were equally distributed on both sides. Most fibers are unambiguously modiolar- or pillar-contacting. In three pillar-contacting SGN, some branches slightly crossed to the modiolar-side yielding polarization vector values that were slightly more positive than −1 (i.e. *Figure 1—figure supplement 1*); however, terminal branches did not equally project to both sides of the inner hair cell.

## Appendix 2

The extended morphology of SGN in the semi-intact preparation raises the possibility of inadequate space-clamp and non-isopotential behavior. This stands in contrast to isolated ganglion preparations where recordings are assumed to come from a spatially compact and isopotential compartment. It was important to carefully evaluate this possibility since non-isopotential behavior would mean that spatial gradients in input conductance (as shown in our results) could come from some unknown combination of variations in the spatial extent of voltage clamp, somatic ion channel densities as well as in the size of currents shunting through the axial conductance and adjacent dendritic compartments (e.g. circuit diagram in *Figure 4—figure supplement 1*.A). To our surprise, SGN in the semi-intact preparation largely behaved like isopotential compartments of about the same size. As a result, we suggest that the gradients we reported here reflect gradients in the ion channel densities on SGN somata.

Our arguments are summarized in the four paragraphs below. First, we exclude the concern about non-isopotential behavior by examining the time-course over which the capacitive transient current decays in voltage clamp. Second, we calculate the size of the 'well-clamped' portion of the membrane by estimating the effective capacitance, also in voltage clamp. Third, we conclude that differences in axial conductance did not produce spatial gradients in input conductance by testing for correlations between dendritic diameter and net-input conductance for Type I fibers. Finally, we exclude the possibility that systematic errors in series resistance could account for the gradients in biophysical properties reported here.

### Isopotential behavior

We tested for non-isopotential behavior by analyzing high resolution recordings of capacitive transient currents ($I_{CV}$) in voltage clamp. These measurements were only made in a subset of recordings. The capacitive transient current was measured by hyperpolarizing the neuron from −60 mV to −65 mV (Figure S.B). This produced a small inward current that decayed. The capacitive portion of the current was isolated by first removing the steady-state current (shaded area under the current). In non-isopotential cells, the decay of the capacitive current can best be described by a sum of exponential decays (e.g. *Equation (7)*).

$$I_{CV} = A_1 e^{-\frac{t}{1}} + A_2 e^{-\frac{t}{2}} + C \tag{7}$$

For example, the capacitive currents in retinal bipolar cells and the Purkinje cells are best fitted with a bi-exponential function, with each exponent term representing the sequential decay across two compartments representing the soma and neurite (*Golowasch et al., 2009*; *Taylor, 2012*; *Rall, 1969*). In contrast, the decay from an iso-potential cell is adequately described by a single exponential term.

Here we fit the capacitive transient currents with *Equation (7)* and compared the coefficients ($A_1$, and $A_2$) of the two exponential terms where $\tau_2 < \tau_1$ (*Figure 4—figure supplement 1*.C). The ratio between the fast and slow terms ($A_2/A_1$) shows that the fast component represented more than 80% of the transient current in 11 out of 16 neurons and was observed in both pillar- and modiolar-contacting SGN (*Figure 4—figure supplement 1*.D). Since the fast component dominates, this suggests that our recordings are largely isopotential.

### Size of well-clamped membrane

We estimated the extent of the plasma membrane clamped under voltage-clamp by computing the effective capacitance (*Taylor, 2012*). Capacitance values averaged at 7.8+/-0.35 pF (n = 35) and were not correlated with normalized basal position (r = 0.05, p=0.85)

(*Figure 4—figure supplement 1*.E). These measurements suggest that there was no systematic modiolar to pillar gradient in the cell surface area clamped by the recording electrode.

Based on confocal scans in the ganglion, we estimate that the SGN in our preparation had an average diameter of around 12.7 +/- 0.3 μm, n = 28. This observation is consistent with measurements made in other mammalian species where somatic diameters range between 10–15 μm (*Tsuji and Liberman, 1997*; *Berglund and Ryugo, 1991*; *Romand and Romand, 1987*; *Nadol et al., 1990*). Assuming an unmyelinated spherical soma (with a specific capacitance around ~1 μF/cm$^2$), the somatic capacitance would be around 5.1 +/- 0.2 pF. Based on the similarities in values between the capacitance as estimated from voltage clamp and from estimated cell diameters, we conclude that we are largely clamping and measuring the currents local to the SGN somata.

In this study, we were not able to measure the extent of the myelin that may be present on the neuron, and therefore cannot assess how differences in myelination and consequent variations in specific capacitance influence our measured capacitance. Because we remove myelinating satellite cells before recording from the SGN soma, we can assume that there is some myelination of the cell bodies and/or neurite. However, we cannot account for SGN having differences in the extent of their myelination which would affect the specific capacitance and total surface area of the neuron. Furthermore, myelination of the cell bodies and/or the neurite may increase the length constant of voltage changes along the neuron and could effectively make the neuron display isopotential behavior. Further studies accessing the passive membrane properties of SGN in the acute semi-intact preparation are needed to determine whether the measured capacitance of the SGN changes when different degrees of myelin are present.

Note that the capacitance or membrane surface area of a passive membranes can also be measured in current-clamp; however, SGN have voltage-dependent conductances that activate near resting potential (~−60 mV). The distortions caused by the activation of these conductances make it difficult to accurately estimate capacitance in current clamp.

## Axial conductance

Differences in biophysical behavior measured in voltage-clamp did not arise from differences in the axial conductance, where neurons with larger diameters would have larger axial conductance. Based on our fiber diameter measurements (*Figure 7*), the axial resistance of type I SGN does not change significantly as SGN contact position moves along the modiolar-pillar axis. In contrast, type II SGN had significantly smaller fiber diameters than type I SGN but had similar net conductance values as that of pillar-contacting type I SGN, suggesting that variations in axial diameter did not translate into significant variations in input conductance.

## Series resistance

Series resistance estimates were not significantly different in SGN along the modiolar-pillar axis, suggesting that there were negligible errors in measuring the magnitude of currents in voltage-clamp. Series resistance estimates of identified type I SGN ranged from 2.6 to 11.0 MΩ with an average of 6.6 +/- 2.3 MΩ (n = 20). This series resistance estimate was measured via the Multiclamp 700B membrane seal test by analyzing the capacitive transient current. In some cases, these estimated series resistances were less than the pipette resistance, which lead to us question the accuracy of the measurement. This possible inaccuracy of estimating the series resistance led to our decision to report our voltage-clamp measurement uncorrected for series resistance. Series resistance estimates as a function of basal position displayed a slight negative linear trend (*Figure 4—figure supplement 1*.F). However, when we performed series resistance corrections on our voltage-clamp measurements, our initial assessment is maintained with maximum magnitudes of steady-state currents significant correlated with basal position ($R^2$ = 0.29, p=0.0003) (*Figure 4—figure supplement 1*.F, red trace). To further assess how much error in the series resistance estimation is needed to

collapse the trends of steady-state currents with basal position, we simulated increasing levels of correlations of the series resistance estimations to the normalized basal position scale (*Figure 4—figure supplement 1*.G, blue and black traces). Our analysis shows that there would need systematic errors six times stronger than the amount we measured in our dataset to collapse these trends (*Figure 4—figure supplement 1*.F, black trace). Because series resistance of a patched SGN was not correlated with whether the neuron contacted the modiolar- or pillar-face of the inner hair cell, magnitude of currents measured in the voltage-clamp were not significantly impacted to cause concern for our biophysical assessment.

Here we did not compensate for or correct voltage errors resulting from the series resistance as we are accustomed to do in isolated neurons. We chose to report raw whole-cell currents to avoid inadvertently adding variability from cell to cell that may result from incorrect estimates of series resistance. Our decision stemmed from our initial concerns that space-clamp errors in the semi-intact preparations would make it difficult to accurately estimate series resistance using the built-in capacitance neutralization function in the Multiclamp 700b. The neutralization procedure assumes a single exponential decay for an iso-potential cell, as would be reasonable for the compact morphology of small isolated somata (see above). We were concerned that imperfect neutralization of capacitance transients would produce errors in both series resistance and membrane capacitance estimates. To avoid adding additional sources of variability to our voltage-clamp estimates of currents, we chose not to apply series resistance corrections or to normalize currents by cell capacitance.

