## [Decision Letter]

**Acceptance summary:**

The mechanosensitive inner hair cells of the inner ear convey their electrical signals to the auditory nerve through synapses at their base. This article shows that important biophysical properties of the auditory-nerve fibres vary systematically with the spatial location of the synapses. It thereby casts light on the organisation of subgroups within the auditory-nerve fibres, and will aid a better understand of the coding of sound in the auditory nerve.

**Decision letter after peer review:**

Thank you for submitting your article "Loudness sensitivity in the spiral ganglion emerges from a maturational gradient in morphology and biophysics" for consideration by *eLife*. Your article has been reviewed by three peer reviewers, including Tobias Reichenbach as the Reviewing Editor and Reviewer #1, and the evaluation has been overseen by Barbara Shinn-Cunningham as the Senior Editor. The following individual involved in review of your submission has agreed to reveal their identity: Elisabeth Glowatzki (Reviewer #2).

The reviewers have discussed the reviews with one another and the Reviewing Editor has drafted this decision to help you prepare a revised submission.

Summary:

The study investigates how biophysical parameters of auditory-nerve fibers vary with the location of the corresponding synapses on the inner hair cells. The authors find that some of these parameters – such as conductances, voltage threshold for spiking and time constants – vary systematically with the spatial location of the synapses. Moreover, they show that a multiple linear regression model that takes these biophysical parameters into account can predict the spatial location to a significant extent. This work is very timely, as the current research in the field focusses on dissecting the underlying properties of auditory-nerve fibre subgroups, an important step in understanding how the sound signal is coded.

Essential revisions:

1) The title and Abstract appear partly misleading. First, the term "loudness sensitivity" in the title leads the reader to expect a direct investigation of the SGNs' dependency of spontaneous rate and threshold on the location. In addition, is "intensity sensitivity" the best descriptor? After all, high spontaneous afferents that respond to low intensity sound do respond to high intensity sound as well. Perhaps "intensity selectivity", "dynamic range" or even "intensity tuning" better captures the difference. Second, the finding of the maturational gradient takes major space in the title and manuscript. This ignores the probably more important finding regarding the pillar/modiolar differences that are found regardless of age. In particular, it is impressive that pillar/modiolar differences already exist at young postnatal ages, which may be worth highlighting. In addition, the authors might want to emphasize their important conclusion that the developmental gradient is not the determinant of the final differentiation of type I cochlear afferents.

2) It is unclear how the authors have dealt with the multiple comparisons that frequently arise. – In “Statistical Analysis”, they write that, when encountering multiple conditions, they have conducted ANOVA followed by post-hoc tests. Did they adjust for multiple comparisons in these post hoc tests? If so, which method did they chose? These issues arise, for instance, on – subsections “Diversity of firing patterns in current clamp”, and – “ Model 1: Spiking neurons only (P3-P10)”.

3) Additional statistical tests are required in some instances:

– Subsection “Current-clamp features of non-spiking neurons are also correlated with normalized basal position” third paragraph: Do the response latencies also vary significantly with position when spiking and non-spiking neutrons are considered separately?

– Subsection “D. Spatial gradients in biophysical properties may be maturational gradients” paragraph one: Please perform a significance test to determine whether the variation in the currents with age is significant.

– In paragraph two: Please perform a significance test for the dependency of the average latencies on age.

– In paragraph three: Please add a significance test for the divergence of the current threshold and the maximal conductance of fibres at the different locations with age.

4) Regarding the linear regression model, the authors determine the variance inflation factor (VIF), and subsequently eliminate variables with a VIF larger than 4. But this is a somewhat arbitrary threshold. The standard and more rigorous way to prevent overfitting is through cross-validation: determine the model parameters from one part of the data (training set) and determine the obtained correlation from the remaining data (test data). Variables can then be eliminated if they improve the model performance (assessed on the test data!). There is also an issue with the model description. Why is the fourth parameter called xn and not x4? This is particularly confusing (and inconsistent) when the text later refers to any variable as xn ( – “Statistical Analysis”). Please clean this up. It should also be stated which variables x1, x2 and so forth represent.

5) The authors say relatively little about how the observed differences in membrane properties contribute to known functional diversity of cochlear afferents. Yes, modiolar afferents have higher current thresholds, but they also have more negative voltage thresholds. Which is more important?

6) In the Discussion, the authors seem to suggest that neurons with "graded firing" may be those that suffered greater damage. Was that the intent?

---

## [Author Response]

Essential revisions:1) The title and Abstract appear partly misleading. First, the term "loudness sensitivity" in the title leads the reader to expect a direct investigation of the SGNs' dependency of spontaneous rate and threshold on the location. In addition, is "intensity sensitivity" the best descriptor? After all, high spontanteous afferents that respond to low intensity sound do respond to high intensity sound as well. Perhaps "intensity selectivity", "dynamic range" or even "intensity tuning" better captures the difference.

The title and Abstract are revised: We appreciate that there is something dissatisfactory about the term sensitivity given that the essential differences across low-thresh/high-SR and high-thresh/low-SR fibers are differences in threshold and absolute range rather than a difference in the slope of the rate-level function. We’ve revised the document to be more careful with the term “loudness-sensitivity”. Instead, we now refer to “intensity” rather than “loudness” and “thresholds” rather than “sensitivity”. We also considered the reviewers’ alternate suggestions… “selectivity” and “tuning”. Both terms left some in-house readers under the impression that the fibers did not respond to intensities outside a particular-intensity range, when in-fact (as the reviewer points out), high-SR units still respond to high-intensities, albeit with much reduced sensitivity. We decided that variations in “threshold” accompanied with “limited range” was perhaps the simplest and most accurate alternative.

Second, the finding of the maturational gradient takes major space in the title and manuscript. This ignores the probably more important finding regarding the pillar/modiolar differences that are found regardless of age. In particular, it is impressive that pillar/modiolar differences already exist at young postnatal ages, which may be worth highlighting. In addition, the authors might want to emphasize their important conclusion that the developmental gradient is not the determinant of the final differentiation of type I cochlear afferents.

Thanks for the pointing out the imbalance in emphasis.

Although we wanted to report on the interesting observation of a local developmental gradient under the IHC, we did not intend to unduly stress this over other results. We’ve significantly revised the text in the Results, Discussion, Abstract, and title to re-balance the overall emphasis placed on this. As recommended, the Discussion and Abstract are now more explicit in stating that spatial gradients are present at both early and late post-natal stages. We also clarify here and in the text that by using the word “maturation” we do not intend to suggest that full maturation means that biophysical gradients will eventually disappear. Rather, our results suggest that the opposite is true. We show that the degree of biophysical heterogeneity is maintained even as the biophysical properties of the SGN population continue to change with post-natal age.

2) It is unclear how the authors have dealt with the multiple comparisons that frequently arise. In “Statistical Analysis”, they write that, when encountering multiple conditions, they have conducted ANOVA followed by post-hoc tests. Did they adjust for multiple comparisons in these post hoc tests? If so, which method did they chose?

We’ve cleaned up several areas where the statistical analysis was inadequately described. This involved expanding the statistical analysis section of the Materials and methods, adding tests of statistical significance as requested by the reviewer, and reporting the results of the tests within the text and/or in Tables 3 and 4.

Figure 2E there are multiple instances where the means of biophysical properties are compared across three firing-pattern groups. We’ve improved the statistical analysis in this figure. Now we use Welch’s ANOVA followed by Games-Howell HSD post-hoc tests to compare the means across the three groups with unequal variance. This is now described in the Materials and methods.

These issues arise, for instance, on subsections “Diversity of firing patterns in current clamp”, and “ Model 1: Spiking neurons only (P3-P10)”.

We are reporting a single regression against normalized basal position. We report Pearson’s correlation coefficient in APA format to show that biophysical properties are significantly correlated along the normalized basal position. Here, although we color code to indicate firing pattern group, we did not intend to compare the regressions between groups. This is because our intention is to move away from the categorical classification based on firing-pattern and move to more quantitative measures for describing biophysical heterogeneity.

3) Additional statistical tests are required in some instances:– Subsection “Current-clamp features of non-spiking neurons are also correlated with normalized basal position” third paragraph: Do the response latencies also vary significantly with position when spiking and non-spiking neutrons are considered separately?– Subsection “D. Spatial gradients in biophysical properties may be maturational gradients” paragraph one: Please perform a significance test to determine whether the variation in the currents with age is significant.– In paragraph two: Please perform a significance test for the dependency of the average latencies on age.– In paragraph three: Please add a significance test for the divergence of the current threshold and the maximal conductance of fibres at the different locations with age.

The reviewers requested several additional significance tests for regressions where we had both continuous and categorical independent variables. In these cases, we performed a 2-way analysis of covariance (ANCOVA). The results of these tests are now included in the text and/or in Table 3.

To address a specific reviewer question…The interpretation of the ANCOVA analysis for latency as a function of position and firing group (spiking vs spiking) is the following: Yes, the latencies vary significantly with position for both spiking and non-spiking groups. However, the latency is not different between the two groups (neither in its mean nor its dependence on basal position). This is now stated in the text.

4) Regarding the linear regression model, the authors determine the variance inflation factor (VIF), and subsequently eliminate variables with a VIF larger than 4. But this is a somewhat arbitrary threshold. The standard and more rigorous way to prevent overfitting is through cross-validation: determine the model parameters from one part of the data (training set) and determine the obtained correlation from the remaining data (test data). Variables can then be eliminated if they improve the model performance (assessed on the test data!).

Yes, the reviewer is correct that one method to prevent overfitting and validate the model would be to reserve data into training and test sets. This sort of cross-validation is especially effective when dealing with large sample sizes. Here we chose VIF as an alternate strategy because pulling appropriate training and test groups is much harder with small data sets. The two dimensions in our data (spatial position and age) make it difficult to confidently reserve a test data-set without either under-representing or over-representing data from particular portions of the two dimensions. The VIF strategy is an alternate strategy for reducing variables to prevent overfitting. It is qualitatively like other variable reduction strategies which apply a weighting factor to penalize redundant variables. Here the penalization is strict in that it completely removes the variable. If anything, the VIF thresholds we set are likely to be overly conservative in eliminating covariate variables, so much so that in some cases our model was reduced to one variable. An incidental benefit of our strategy is that we were explicitly examining the covariations across variables at every stage of the backward stepping procedure. This helped us to easily recognize the relationships across variables and to quickly home in on a biophysically interpretable covariation between conductance, current threshold and latency.

We also tried cross-validation (like that suggested by the reviewer) by reducing the resolution across one dimension of the data. We did this by collapsing the continuous variable for spatial position into a bi-modal classification (modiolar/pillar). This allowed us to reserve 75% of the data into a training data set and 25% of the data into a test data set. In a preliminary three-fold validation, a logistic regression model based on the training data set was successful at predicting spatial position (modiolar versus pillar) in the test data set with 75% accuracy. However, we did not push this analysis approach because it dilutes one part of our message, namely, that the SGN biophysics form a continuum rather than falling into distinct subgroups.

There is also an issue with the model description. Why is the fourth parameter called xn and not x4? This is particularly confusing (and inconsistent) when the text later refers to any variable as xn (“Statistical Analysis”). Please clean this up. It should also be stated which variables x1, x2 and so forth represent.

We’ve cleaned up the equations to indicate that x1, x2,…xn are a sequence of variables. Note that for each rendition of the model, the order of x1 through xn change according to the influence of the variable on the model. The variables corresponding to x1, x2 and so forth are shown in the tables. We’ve now added a pointer in the Model Building section in the Materials and methods to indicate this.

5) The authors say relatively little about how the observed differences in membrane properties contribute to known functional diversity of cochlear afferents. Yes, modiolar afferents have higher current thresholds, but they also have more negative voltage thresholds. Which is more important?

We’ve now expanded the discussion under a sub-heading (relating membrane properties to cochlear afferent function) to spell out our thinking.

Regarding the specific question about current versus voltage thresholds. Our description of voltage threshold in the text was incomplete. We’ve revised the text related to Figure 2 to make the following clarification. Both voltage-thresholds and resting potential change significantly with neurobasal position (the correlations are in Table 1). As a result, the change in relative threshold with neurobasal position is not significant (r(20)=0.29, p=0.20). In other words, in order to spike, pillar and modiolar-contacting SGN need to experience a similar depolarization relative to resting potential. However, from the stimulus perspective, our results show that larger currents are needed to affect a similar degree of depolarization in modiolar-contacting SGN. In summary, the important distinction between modiolar- and pillar-contacting SGN revealed here is the difference in current threshold rather than in voltage threshold. This is now reported in the text.

How might such differences in current threshold contribute to well the known functional diversity of cochlear afferents? We’ve added to the Discussion where we layout our thinking. Briefly, differences in current threshold and temporal integration time (as implied by first-spike latency) could make some neurons responsive only to subtypes of EPSCs. Since EPSCs at auditory synapses are heterogenous in amplitude and kinetics, two neurons could respond to the same EPSC train differently because of variation in their intrinsic properties. These ideas are now proposed in the Discussion.

6) In the Discussion, the authors seem to suggest that neurons with "graded firing" may be those that suffered greater damage. Was that the intent?

Thank you for pointing this out. We’ve re-written that section to clarify.

Our intent was not to suggest that these neurons are damaged since the recordings were made after giga Ohm seals, had stable resting potentials and voltage-clamp responses. Also, many of the subthreshold features of these neurons were indistinguishable from that of spiking-cells, thereby allowing us to pool the two groups together. That their peripheral dendrites were robustly filled with biocytin is another indication that the cells were healthy…in our experience, unhealthy cells did not fill properly. Since ion channels don’t easily insert under myelin, it is possible that the graded-firing neurons reflect a true difference in somatic myelination and the spatial distribution of channels along the neuron. In contrast, SGN in cultured preparations tend to demyelinate over a longer period. Maybe ion channels are more successfully targeted to the cell body after disassociating and culturing. However, we cannot definitively eliminate the possibility that demyelination causes some unknown damage to the neurons, but we think otherwise based on the above arguments.